# DANCE provides an open-source and low-cost approach to quantify aggression and courtship in *Drosophila*

R Sai Prathap Yadav[1†], Paulami Dey[1†‡], Faizah Ansari[1], Tanvi Kottat[1], Manohar Vasam[1], P Pallavi Prabhu[1], Shrinivas Ayyangar[1], Swathi Bhaskar S[1], Krishnananda Prabhu[2], Monalisa Ghosh[1§], Pavan Agrawal[1*]

[1]Centre for Molecular Neurosciences, Kasturba Medical College, Manipal Academy of Higher Education, Manipal, India; [2]Department of Biochemistry, Kasturba Medical College, Manipal Academy of Higher Education, Manipal, India

*For correspondence:
pavan.agrawal@manipal.edu

[†]These authors contributed equally to this work

Present address: [‡]Institute for Developmental Biology and Neurobiology, Johannes Gutenberg University Mainz, Biozentrum I, Mainz, Germany; [§]Gottfried Schatz Research Center, Molecular Biology and Biochemistry, Medical University of Graz, Graz, Austria

## eLife Assessment

This study presents a **valuable** open-source and cost-effective method for automating the quantification of male aggression and courtship in *Drosophila melanogaster*. The work as presented provides **solid** evidence that the use of the behavioral setup that the authors designed - using readily available laboratory equipment and standardised high-performing classifiers they developed using existing software packages - accurately and reliably characterises social behavior in Drosophila. The work will be of interest to Drosophila neurobiologists and particularly to those working on male social behaviors.

**Abstract** Quantifying animal behavior is pivotal for identifying the neuronal and genetic mechanisms involved. Computational approaches have enabled automated analysis of complex behaviors such as aggression and courtship in *Drosophila*. However, existing approaches rely on rule-based algorithms and expensive hardware, limiting sensitivity to behavioral variations and accessibility. Here, we present the <u>D</u>rosophila <u>A</u>ggression a<u>n</u>d <u>C</u>ourtship <u>E</u>valuator (DANCE), a low-cost, open-source platform that combines machine learning-based classifiers and inexpensive hardware to quantify aggression and courtship. DANCE consists of six novel behavioral classifiers trained using a supervised machine learning algorithm. DANCE classifiers address key limitations of rule-based algorithms, capturing dynamic behavioral variations more effectively. DANCE hardware is constructed using medicine blister packs and acrylic sheets, with recordings acquired using smartphones, making it affordable and accessible. Benchmarking demonstrated that DANCE hardware performs comparably to high-cost setups. We validated DANCE in diverse contexts, including social isolation vs. enrichment, which modulates aggression and courtship; RNAi-mediated downregulation of the neuropeptide Dsk; and optogenetic silencing of dopaminergic neurons, which promotes aggression. DANCE provides a cost-effective and portable solution for studying behaviors in resource-limited settings or near natural habitats. Its accessibility and robust performance democratize behavioral neuroscience, enabling rapid screening of genes and neuronal circuits underlying complex social behaviors.

## Introduction

Detailed and accurate annotation and analysis of complex behaviors are necessary for understanding the underlying neural and molecular mechanisms. The fruit fly *Drosophila melanogaster* is one of

the most accessible and well-studied model organisms for identifying the neuronal and molecular underpinnings of behavior. Multiple large-scale screens have been conducted in *Drosophila* to study complex social behaviors such as aggression and courtship (*Asahina, 2017*; *Greenspan and Ferveur, 2000*; *Hall, 2002*; *Kravitz and Fernandez, 2015*) to identify the underlying neural circuitry (*Agrawal et al., 2020*; *Asahina et al., 2014*; *Davis et al., 2018*; *Hoopfer et al., 2015*; *Yadav et al., 2024*) and genes involved (*Agrawal et al., 2020*; *Benzer, 1967*; *Gill, 1963*; *Hall, 1978*; *Ishii et al., 2022*; *Wang et al., 2008*). These behaviors exhibit distinct, stereotyped patterns. For example, aggression involves chasing, fencing (*Jacobs, 1960*), wing threats, boxing (*Dow and von Schilcher, 1975*), lunging, and tussling (*Hoffmann, 1987a*; *Hoffmann, 1987b*). Similarly, courtship consists of multiple stereotyped behaviors exhibited by the male fly, such as orienting, circling, and following the female (*Cook and Cook, 1975*; *Markow, 1987*; *O'Dell, 2003*). To stimulate the female to be more receptive, the male produces a species-specific song by vibrating and extending its wing (*Bennet-Clark and Ewing, 1969*; *Swain and von Philipsborn, 2021*). The male then attempts copulation by curling its abdomen and finally mounts the female for copulation (*Bastock and Manning, 1955*; *Spieth, 1974*).

Manual analysis by trained observers is considered the gold standard in behavioral analysis, but it is time-consuming and unsuitable for large-scale screens (*Gomez-Marin et al., 2014*; *Robie et al., 2017a*). 'Computational ethology' (*Anderson and Perona, 2014*; *Datta et al., 2019*) helps address this challenge by automating behavioral annotation by leveraging advances in computer vision and machine learning (*Robie et al., 2017b*). This enables high-throughput behavioral screening to identify responsible genes and circuits.

A typical computational ethology workflow involves recording animal behaviors and tracking their positions along with body movements. This is followed by the analysis and classification of the observed behaviors from hundreds to thousands of video frames capturing behavioral instances. Several software programs, such as Ctrax, Caltech FlyTracker, and Deep Lab Cut (*Branson et al., 2009*; *Eyjolfsdottir et al., 2014*; *Mathis et al., 2018*), are widely used for tracking behaviors in *Drosophila*. Each comes with strengths and weaknesses. Ctrax (*Branson et al., 2009*) can accurately track fly position and movement, but identity switches remain a challenge, especially when tracking groups of flies. While both Ctrax and FlyTracker (*Eyjolfsdottir et al., 2014*) may produce identity switches, when groups of flies were tracked simultaneously, Ctrax led to inaccuracies that required manual correction using specialized algorithms such as FixTrax (*Bentzur et al., 2021*).

The effectiveness of various machine learning pipelines is eventually measured by comparing their output to human annotation, called 'ground-truthing'. A rule-based algorithm such as CADABRA (*Dankert et al., 2009*) is used to quantify aggression, but it can lead to mis-scoring and identity switches, as revealed by ground-truthing (*Simon and Heberlein, 2020*), which needs to be corrected in a semiautomated manner (*Kim et al., 2018*). MateBook (*Ribeiro et al., 2018*) is another rule-based algorithm used to quantify courtship; however, similar to CADABRA, it tends to miss true-positive events, leading to significant mis-scoring of behaviors under certain experimental conditions.

The Janelia Automatic Animal Behavior Annotator (JAABA) (*Kabra et al., 2013*) addresses the challenges of rigid rule-based approaches by employing a supervised learning approach. In the JAABA pipeline, user-labeled data are utilized for training to encompass the dynamic variations in behaviors, allowing it to predict behaviors on the basis of learning from input data.

Several studies have developed JAABA-based behavioral classifiers for measuring aggression (*Chiu et al., 2021*; *Chowdhury et al., 2021*; *Duistermars et al., 2018*; *Leng et al., 2020*; *Tao et al., 2024*) and courtship (*GilMartí et al., 2023*; *Pantalia et al., 2023*). However, many of these studies did not make these classifiers publicly available (*Duistermars et al., 2018*; *GilMartí et al., 2023*; *Pantalia et al., 2023*). In other cases, the reported approaches relied on specialized hardware, such as custom 3D-printed parts (*Chowdhury et al., 2021*; *GilMartí et al., 2023*), or high-end machine-vision cameras (*Chiu et al., 2021*; *Chowdhury et al., 2021*; *Duistermars et al., 2018*; *Hindmarsh Sten et al., 2025*; *Leng et al., 2020*; *Tao et al., 2024*), limiting their accessibility and wider adoption.

Here, we describe DANCE (*D*rosophila *A*ggression an*d* *C*ourtship *E*valuator), an open-source, user-friendly analysis and hardware pipeline to simplify and automate the process of robustly quantifying aggression and courtship behaviors. DANCE has two components: (1) A set of robust, machine vision-based behavioral classifiers developed using JAABA to quantify aggression and courtship. (2) An inexpensive hardware setup built from off-the-shelf materials and consumer smartphones for behavioral recording. Compared with previous methods (*Dankert et al., 2009*; *Ribeiro et al., 2018*), the

DANCE classifiers improved accuracy and reliability, while its low-cost hardware eliminates the need for specialized arenas and cameras. All classifiers and analysis codes are publicly available, enabling broad adoption, especially in resource-limited settings. Together, DANCE provides a powerful, accessible platform for behavioral screening and the discovery of mechanisms underlying complex social behaviors and neurological disorders.

## Results

### DANCE assay analysis pipeline

To overcome the challenge of time-consuming manual behavioral annotation or resource-intensive, complex hardware, we developed an automated, high-throughput quantification pipeline—DANCE and trained new behavioral classifiers using an existing machine learning algorithm, JAABA (*Kabra et al., 2013*)—to robustly quantify aggression and courtship in *Drosophila* (*Figure 1*). We also designed a simple, low-cost recording setup constructed from repurposed transparent medicine blister packs, acrylic sheets, and paper tape, enabling easy behavioral recordings. To record these behaviors, we used Android smartphone cameras and an electronic tablet or smartphone serving as a backlight illumination source (*Figure 1A*, Materials and methods).

We quantify these behaviors using our DANCE classifiers. Unlike existing setups that cost approximately USD 3500, DANCE hardware can be assembled from off-the-shelf components for less than USD 0.30 (*Supplementary file 1*). To benchmark the performance of the DANCE classifiers, we used pre-existing setups and rule-based methods for quantifying courtship and aggression and compared their performance with that of the DANCE classifiers (*Figure 1B and C*).

To train the DANCE classifiers using JAABA (*Kabra et al., 2013*), for aggressive lunges, we used an existing setup described in *Dankert et al., 2009*, modified from *Dierick, 2007*; *Figure 1A*; *Figure 1—figure supplement 1*, and for courtship behaviors, we used a pre-existing setup described in *Koemans et al., 2017*; *Figure 1A*; *Figure 1—figure supplement 2*. We tracked the position, motion, and interactions of pairs of flies across video frames using the Caltech FlyTracker (*Eyjolfsdottir et al., 2014*). To avoid data leakage, we randomly divided the acquired videos into two categories, 'training videos' and 'test videos', to train and evaluate the DANCE classifiers. These test videos were also manually 'ground-truthed' frame by frame, which is considered the gold standard for behavioral annotation (*Figure 1B, D, and E*).

We benchmarked the performance of the DANCE classifiers against existing rule-based algorithms, CADABRA (*Dankert et al., 2009*) and MateBook (*Ribeiro et al., 2018*), and an existing JAABA aggression classifier (*Chowdhury et al., 2021*). Comparisons with manual ground-truth data revealed that the performance of the DANCE classifiers is comparable to that of human annotations and has higher sensitivity than rule-based algorithms (*Figure 1D and E*). The subsequent sections describe the quantitative analysis of individual DANCE classifiers and benchmarking of DANCE hardware.

### DANCE lunge classifier to quantify aggressive behavior

Aggression is an innate, complex behavior, and *Drosophila* males exhibit several stereotyped behavioral patterns during aggressive encounters, with lunging used widely as a measure of overall aggression in males (*Agrawal et al., 2020*; *Asahina et al., 2014*; *Chiu et al., 2021*; *Chowdhury et al., 2021*; *Davis et al., 2014*; *Dierick, 2007*; *Hoffmann, 1987a*; *Hoopfer et al., 2015*; *Hoyer et al., 2008*; *Jung et al., 2020*; *Nilsen et al., 2004*; *Watanabe et al., 2017*; *Yadav et al., 2024*). A lunge is defined as a male fly raising its front legs and hitting down on the other fly.

We developed a new classifier using JAABA (*Kabra et al., 2013*) to robustly quantify aggressive lunges in *Drosophila*, hereafter referred to as the DANCE lunge classifier. We quantified lunges using our classifier from 20-min-long videos and compared the output with manual ground-truth and existing methods—CADABRA (*Dankert et al., 2009*) and the Divider assay classifier (*Chowdhury et al., 2021*). CADABRA tends to miss several true-positive lunges, likely because of its rigid, rule-based framework, which cannot adapt to the dynamic variations in behavior. *Figure 2A* shows the lunge scores from 40 different videos using the ground-truth, the DANCE lunge classifier, CADABRA, and the Divider assay classifier. While the ground-truth and DANCE classifiers' outputs are comparable, CADABRA and the Divider assay classifier underscore lunges across videos. We ground-truthed the DANCE lunge classifier against 40 'test videos' (*Figure 2A*, Materials and methods). Inter-observer

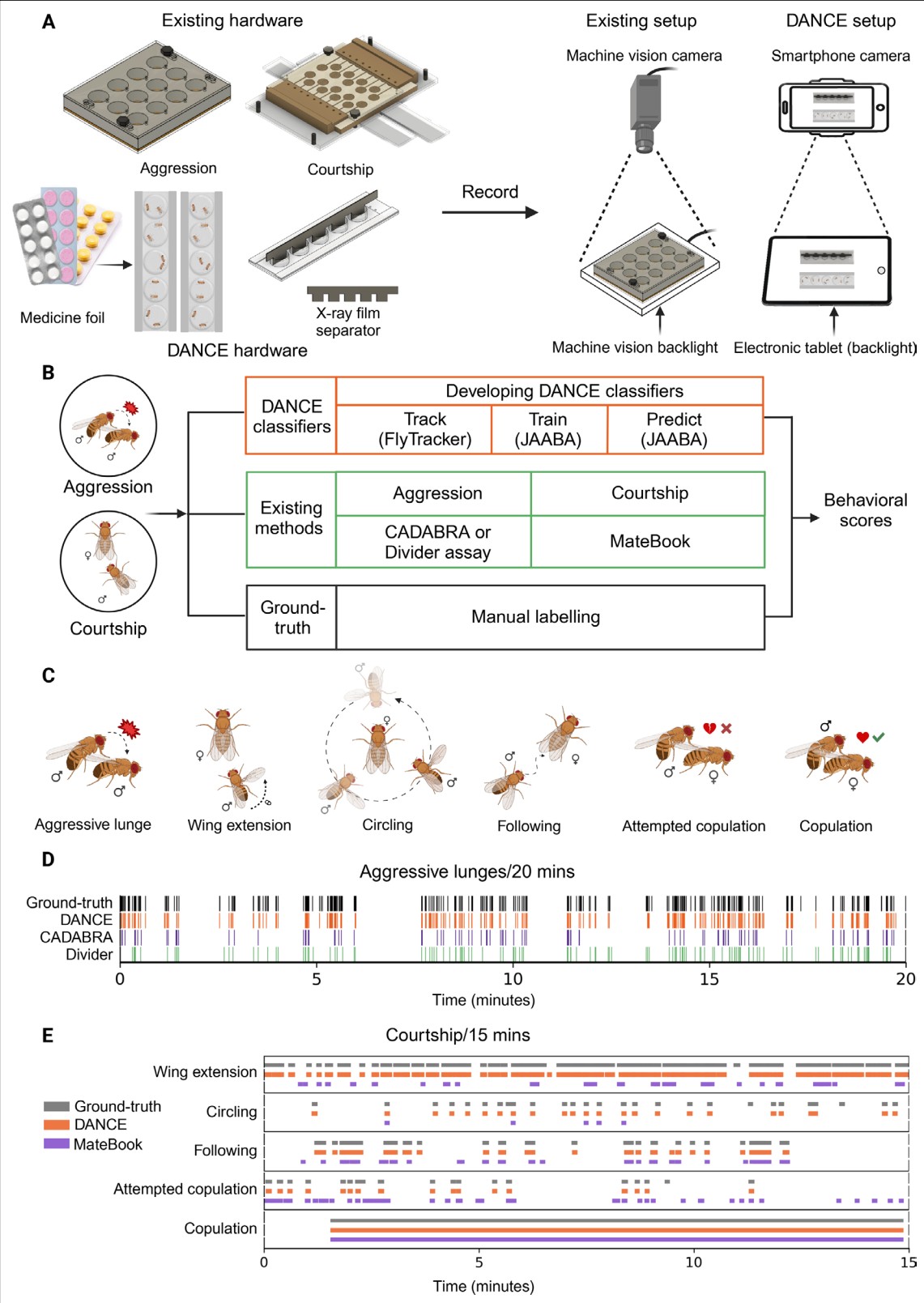

**Figure 1.** The <u>D</u>rosophila <u>A</u>ggression an<u>d</u> <u>C</u>ourtship <u>E</u>valuator (DANCE) assay provides an accessible approach for quantifying aggression and courtship behaviors. (**A**) Comparison of existing machine-vision camera hardware (***Dankert et al., 2009***; ***Koemans et al., 2017***) with the simplified, low-cost DANCE hardware for behavior acquisition. (**B**) Workflow for developing DANCE classifiers, including training, benchmarking against existing methods and manual ground-truth annotations to generate behavioral scores. (**C**) Behavioral classifiers developed to quantify male aggression (lunge) and

*Figure 1 continued on next page*

*Figure 1 continued*

courtship (wing extension, circling, following, attempted copulation, and copulation). (**D**) Representative raster plots comparing ground-truth, DANCE, CADABRA, and Divider assay performance for aggression. (**E**) Representative raster plots comparing ground-truth, DANCE, and MateBook performance for courtship. Created in BioRender.

The online version of this article includes the following figure supplement(s) for figure 1:

**Figure supplement 1.** Aggression chamber described by *Dankert et al., 2009*.

**Figure supplement 2.** Courtship setup described by *Koemans et al., 2017*.

validation confirmed that there were no significant differences between the two independent manual annotations (*Figure 2—figure supplement 1A*).

Since aggressive lunges have a large dynamic range, we benchmarked our classifier across a range of aggressive behaviors and subdivided the ground-truth videos into four categories: (1) low aggressive, 0–70 lunges (*Figure 2B*); (2) moderately aggressive, 71–160 lunges (*Figure 2C*); (3) highly aggressive, 161–300 lunges (*Figure 2D*); and (4) hyperaggressive, >300 lunges (*Figure 2E*). DANCE scores remained comparable to ground-truth scores across all categories, whereas CADABRA and Divider underestimated the lunge counts (*Figure 2B–E*). Correlation analysis revealed a strong relationship between DANCE and ground-truth scores (*Figure 2F*, *Supplementary file 2*). In comparison, CADABRA and the Divider assay classifier showed a weaker correlation (*Figure 2G–H*, *Supplementary file 2*). We reasoned that CADABRA's lower performance is most likely due to the rigid rules used to define a lunge (*Dankert et al., 2009*), whereas the Divider assay classifier, although also JAABA-based, was trained using data from a rectangular arena. Because JAABA classifiers rely on features influenced by the arena geometry, this mismatch likely reduced its accuracy in our circular setup. To further evaluate the performance, we computed the precision, recall, and F1 score (*Figure 2I*). The DANCE lunge classifier achieved a precision of 78.7%, recall of 73.1%, and an overall F1 score of 75.8%, exceeding the values obtained with other methods. Classifier robustness across multiple training videos, including the dataset used for inter-method comparisons (Video 9), is summarized in *Figure 2—figure supplement 2*. Together, our analysis suggests that the DANCE lunge classifier performs with high precision and quantifies lunge numbers robustly over a broad range of fighting intensities.

## DANCE classifiers to quantify courtship behaviors in *Drosophila*

The first report of *Drosophila* courtship behavior described stereotypic behaviors such as wing 'scissor-like' movements, with males 'swaying around the female', licking, tapping, and mounting (*Sturtevant, 1915*). By the 2000s, studies revealed genes and neural circuits involved in courtship (*Dickson, 2008*; *Pavlou and Goodwin, 2013*). Automated analysis techniques for courtship exist, but their adoption has been limited by expensive hardware, reliance on custom parts, or a lack of publicly available code and classifiers (*Duistermars et al., 2018*; *GilMartí et al., 2023*; *Koemans et al., 2017*; *Reza et al., 2013*; *Tao et al., 2024*).

MateBook is a recent rule-based pipeline for automating the quantification of courtship behavior (*Ribeiro et al., 2018*). It relies on predefined rules derived from CADABRA (*Dankert et al., 2009*). To resolve ambiguities in the two overlapping flies when their trajectories are estimated from the video recordings, identities are assigned by relying on the distinct body sizes of the male and female flies, as the females are larger than the males. This size assumption is problematic in assays that use decapitated virgin females, which approximate male size. Such conditions are often used to assess male courtship independent of female behavioral feedback, e.g., when evaluating pheromone effects (*Cook and Cook, 1975*; *Spieth, 1966*). In these contexts, rule-based approaches can introduce false positives and false negatives because the body-size criterion is not met.

To overcome these limitations, we trained and validated five new courtship classifiers using JAABA (*Kabra et al., 2013*). These classifiers quantify distinct stages of the male courtship ritual (*Sokolowski, 2001*), including wing extension, following, circling, attempted copulation, and copulation. To ensure robustness across conditions, training datasets included videos with both decapitated and intact females (mated or virgin). We evaluated the DANCE classifier performance by comparing the outputs with the manual ground-truth and MateBook results. Because courtship behaviors differ in duration, we calculated a behavior index to enable direct comparisons between methods. Details of the

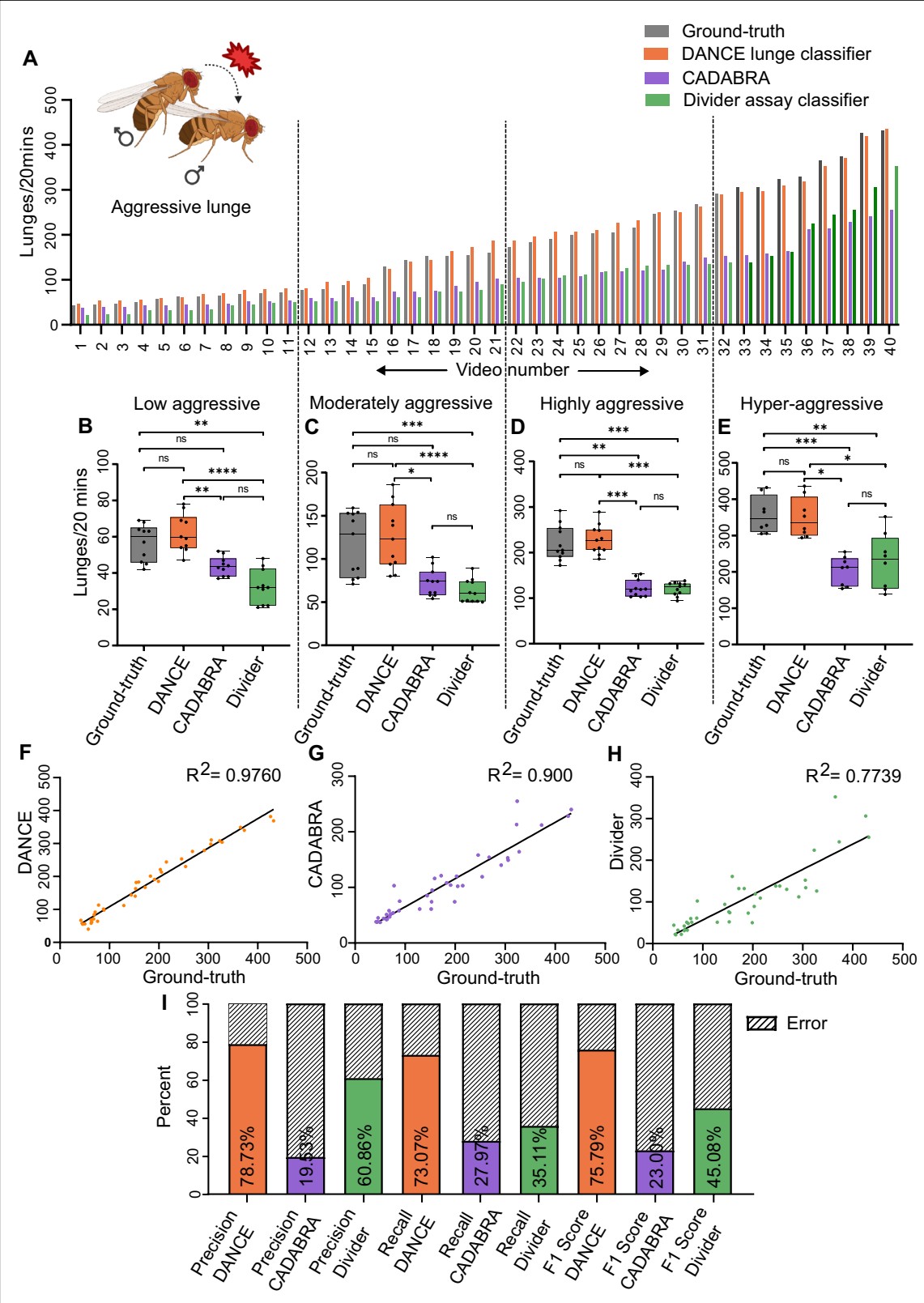

**Figure 2.** Validation of the <u>D</u>rosophila <u>A</u>ggression a<u>n</u>d <u>C</u>ourtship <u>E</u>valuator (DANCE) lunge classifier for quantifying male aggression. (**A**) Lunge scores from 20-min-long videos scored using ground-truth annotations (gray), the DANCE lunge classifier (orange), CADABRA (purple), and Divider assay classifier (green). (**B–E**) Comparison of lunge scores across different aggression levels, based on manual scoring and predictions from DANCE, CADABRA, and Divider: (**B**) 0–70 lunges 'low aggressive' (n=10; ground-truth vs. DANCE ns, p>0.9951, ground-truth vs. CADABRA ns, p>0.3405,

*Figure 2 continued on next page*

*Figure 2 continued*

ground-truth vs. Divider assay classifier **p<0.0017, DANCE vs. CADABRA **p<0.0060, DANCE vs. Divider assay classifier ****p<0.0001, CADABRA vs. Divider assay classifier ns, p>0.4996). (**C**) 71–160 lunges, 'moderately aggressive' (n=11; ground-truth vs. DANCE ns, p>0.9999, ground-truth vs. CADABRA ns, p>0.1247, ground-truth vs. Divider assay classifier ***p<0.0002, DANCE vs. CADABRA *p<0.0102, DANCE vs. Divider assay classifier ****p<0.0001, CADABRA vs. Divider assay classifier ns, p>0.4157). (**D**) 161–300 lunges, 'highly aggressive' (n=11; ground-truth vs. DANCE ns, p>0.9999, ground-truth vs. CADABRA, **p<0.0057, ground-truth vs. Divider assay classifier *p<0.0102, DANCE vs. CADABRA ***p<0.0002, DANCE vs. Divider assay classifier ***p<0.0004, CADABRA vs. Divider assay classifier ns, p>0.9999), and (**E**) >300 lunges, 'hyper-aggressive' (n=8; ground-truth vs. DANCE ns, p>0.9999, ground-truth vs. CADABRA, ***p<0.0006, ground-truth vs. Divider assay classifier **p<0.0029, DANCE vs. CADABRA, *p<0.0402, DANCE vs. Divider assay classifier *p<0.0102, CADABRA vs. Divider assay classifier ns, p>0.9999; Friedman's ANOVA with Dunn's test). (**F**) Regression analysis of the DANCE 'lunge classifier' vs. manual scores ($R^2$=0.9760, n=40). (**G**) Regression of the CADABRA vs. the DANCE lunge classifier ($R^2$=0.9, n=40). (**H**) Regression of the Divider assay lunge classifier score vs. manual score ($R^2$=0.7739, n=40). (**I**) Precision, recall, and F1 scores of the DANCE lunge classifier compared with those of CADABRA and Divider.

The online version of this article includes the following source data and figure supplement(s) for figure 2:

**Source data 1.** Source data for *Figure 2* showing quantitative aggressive lunge counts and performance metrics for DANCE and existing methods used to quantify aggressive behavior in male flies.

**Figure supplement 1.** Comparison of annotations by two independent evaluators to assess observer bias during ground-truthing.

**Figure supplement 1—source data 1.** Source data for *Figure 2—figure supplement 1* comparing behavioral annotations across multiple behaviors (lunges, wing extension, attempted copulation, following, circling, and copulation indices) independently scored by two observers to assess observer bias during ground-truthing.

**Figure supplement 2.** Evaluation of the D̲rosophila A̲ggression an̲d C̲ourtship E̲valuator (DANCE) lunge classifier predictions across training videos.

**Figure supplement 2—source data 1.** Source data for *Figure 2—figure supplement 2* comparing aggressive lunge metrics and performance of different classifiers across training videos to assess robustness and reproducibility.

classifier thresholds and criteria are provided in the Materials and methods and in *Supplementary file 3*. To assess observer bias, annotations from two independent evaluators were compared, revealing no significant differences (*Figure 2—figure supplement 1B–F*, *Supplementary file 4*).

## Wing extension

During unilateral wing extension, a male vibrates its wing at a specific frequency to produce a species-specific courtship song to attract the female (*Shorey, 1962*; *Spieth, 1952*).

*Figure 3A* shows wing extension indices derived from manual ground-truth (gray) and the DANCE wing extension classifier (orange) across 15 videos with decapitated virgin females; the two measures are comparable, whereas MateBook (purple) systematically reports lower scores in most videos (*Figure 3A*). The DANCE wing extension index is strongly correlated with the ground-truth scores (*Figure 3B and C*) but weakly correlated with MateBook (*Figure 3B and D*). The classifier performance metrics confirm high reliability (precision 92.2%, recall 98.1%, F1 95.1%) (*Figure 3E*). Similar trends were observed in the mated female dataset (*Figure 3—figure supplement 1*).

## Attempted copulation and copulation

Copulation typically lasts for approximately 15–25 min, and its duration is primarily determined by the male (*MacBean and Parsons, 1967*). Interrupted mating experiments have shown that sperm are transferred several minutes after copulation begins (*Fowler, 1973*; *Tompkins et al., 1980*). This distinction separates mounting into two outcomes—successful copulation and unsuccessful attempted copulation—for which we developed separate classifiers (see Materials and methods).

The DANCE attempted copulation classifier closely matched the ground-truth across videos (*Figure 4A and B*). Compared with MateBook, which often overestimates attempted copulation events, the DANCE classifier provides more consistent detection. The correlation with the ground-truth was stronger for DANCE as compared to MateBook (*Figure 4C and D*), and the performance metrics for DANCE were robust (precision 82.6%, recall 89.2%, F1 85.8%; *Figure 4E*).

For copulation, which is trained on videos using mated and decapitated virgin females, the DANCE copulation classifier also matches the ground-truth with near-perfect performance (*Figure 4—figure supplement 1*). MateBook performs reasonably well for copulation, but alignment or arena-detection errors in some recordings cause occasional false negatives or positives (*Figure 4—figure supplement 1A*, videos 13 and 18).

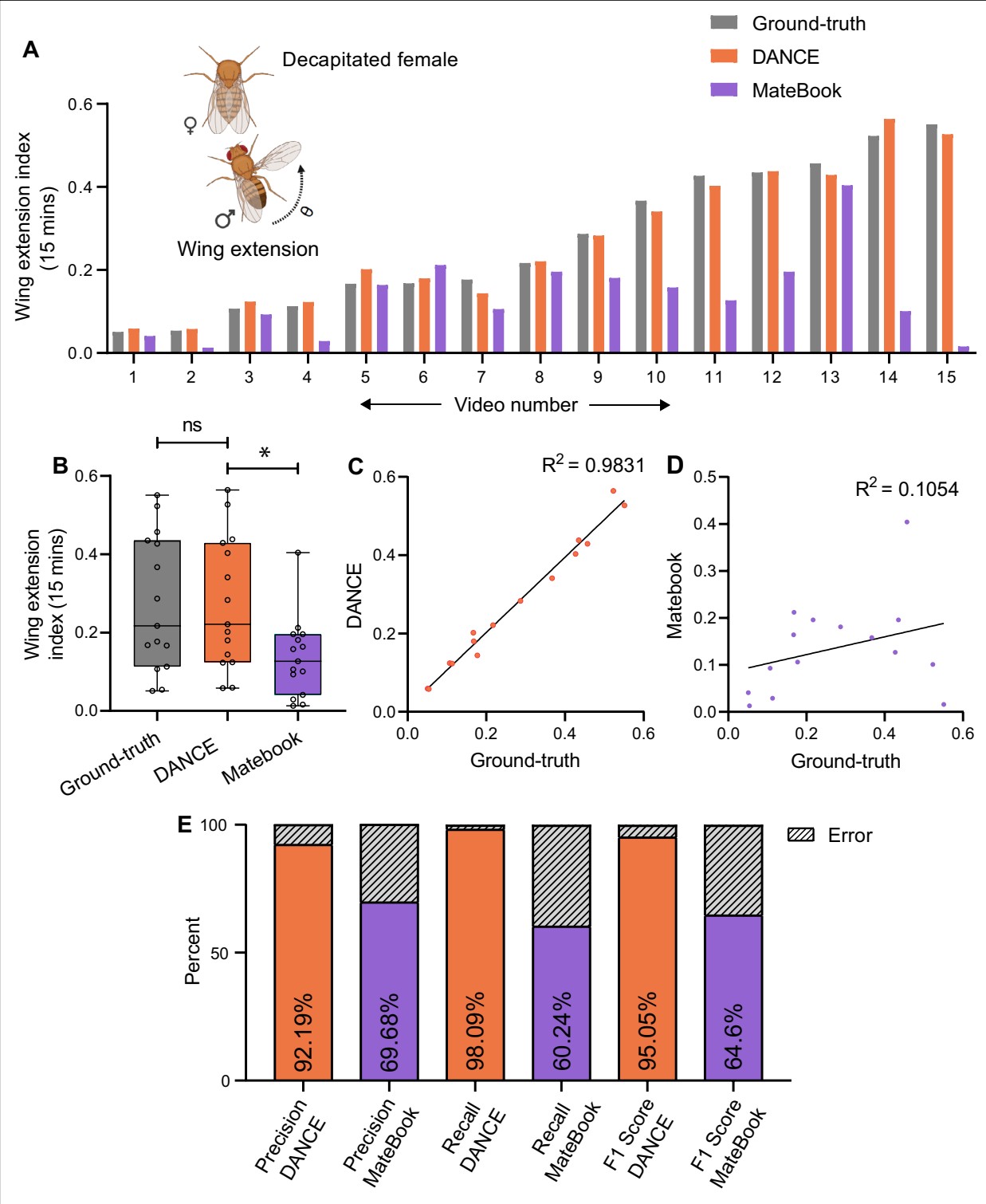

**Figure 3.** Evaluation of the Drosophila Aggression and Courtship Evaluator (DANCE) wing extension classifier for quantifying courtship behavior.
(**A**) Wing extension index of males from 15-min-long videos scored using manual ground-truth annotations (gray), the DANCE wing extension classifier (orange), and MateBook (purple), with decapitated virgin females. MateBook underscored wing extension across multiple videos (Friedman's ANOVA with Dunn's test: ground-truth vs. MateBook **p=0.0020, ground-truth vs. DANCE ns, p>0.9999; n=15). (**B**) Comparison of ground-truth, DANCE, and MateBook wing extension scores (Kruskal–Wallis ANOVA with Dunn's test, ground-truth vs. DANCE ns, p>0.9999, ground-truth vs. MateBook *p=0.0436; n=15). (**C**) Regression analysis of the DANCE wing extension classifier vs. ground-truth ($R^2$=0.9831, n=15). (**D**) Regression of MateBook vs. ground-truth ($R^2$=0.1054, n=15). (**E**) Precision, recall, and F1 score of the DANCE wing extension classifier and MateBook relative ground-truth scores.

*Figure 3 continued on next page*

*Figure 3 continued*

The online version of this article includes the following source data and figure supplement(s) for figure 3:

**Source data 1.** Source data for *Figure 3* showing quantitative behavioral indices and performance metrics (bout-level analysis) for DANCE and existing methods used to quantify wing-extension behavior in decapitated virgin female flies.

**Figure supplement 1.** Evaluation of the D̲rosophila A̲ggression an̲d C̲ourtship E̲valuator (DANCE) wing extension classifier in the mated-female dataset.

**Figure supplement 1—source data 1.** Source data for *Figure 3—figure supplement 1* showing quantitative behavioral indices and performance metrics (bout-level analysis) for DANCE and existing methods used to quantify wing-extension behavior in mated female flies.

## Circling

A male circling around a female is a distinct courtship element implicated in female re-stimulation (*Kessler, 1962*). Variations in circling frequency across species contribute to reproductive isolation (*Brown, 1965*).

We evaluated the DANCE circling classifier on both the decapitated-virgin and mated-female datasets (*Figure 5*, *Figure 5—figure supplement 1*). The circling indices from ground-truth and DANCE were comparable across videos (*Figure 5A and B*), whereas MateBook often under-represented circling in individual recordings (*Figure 5A*). The DANCE circling index correlated strongly with ground-truth scores as compared to MateBook (*Figure 5C and D*), and classifier metrics indicated robust performance (precision 98.0%, recall 92.1%, F1 95.0%; *Figure 5E*).

## Following

During following, the male tracks the female's movement to initiate subsequent courtship acts (*Spieth, 1968*). Because following is a relatively continuous and readily defined behavior, both the MateBook and the DANCE following classifier performed well (*Figure 5—figure supplement 2A–D*). However, the DANCE following classifier produced more balanced scores (precision 91.2%, recall 91.1%, F1 91.1%) compared with MateBook (precision 65.8%, recall 83.4%, F1 73.5%; *Figure 5—figure supplement 2E*), indicating lower rates of false positives and false negatives for DANCE.

Finally, we performed frame-level analyses in addition to bout-level evaluations to provide a more granular assessment of the courtship classifiers (see Materials and methods). Frame-level metrics showed only marginal reductions in performance compared with bout-level metrics (*Figure 5—figure supplement 3*). Together, these results demonstrate that the DANCE classifiers provide a reliable and accurate means to quantify both aggression and courtship behaviors, supporting subsequent benchmarking using the DANCE hardware.

## DANCE hardware

Existing setups for recording *Drosophila* aggression and courtship (*Figure 1A*; *Figure 1—figure supplements 1 and 2*) present several practical challenges that limit their broad adoption. These include the need for complex, custom-fabricated components, 3D-printed parts, specialized machine-vision cameras and backlights, and considerable technical expertise for data acquisition and processing (*Chowdhury et al., 2021*; *Dankert et al., 2009*; *GilMartí et al., 2023*; *Koemans et al., 2017*). Setting up some aggression assays (*Dankert et al., 2009*; *Dierick, 2007*) also requires coating chambers with fluon to prevent flies from walking on the walls, which is labor-intensive.

To provide a low-cost, easy-to-assemble alternative, we developed the DANCE hardware (*Figure 6*), an inexpensive, scalable, and robust system for recording *Drosophila* aggression and courtship behaviors.

The DANCE hardware consists of readily available off-the-shelf components, including transparent medicine blister packs (tablet foils) used as recording chambers, which are mounted on 2 mm acrylic base plates and secured with paper tape (*Figure 6A–D*, *Figure 6—video 1; Figure 6—video 2*). Instead of using machine-vision cameras, DANCE employs widely available Android smartphones for recording and substitutes backlights with tablets or smartphones displaying a white screen to provide uniform illumination (*Figure 6E–G*). Aggression and courtship behaviors were recorded at 30 fps and 1080p resolution.

For the aggression assays (*Figure 6A–B and E*), the blister foil was slid over a base plate containing an apple–juice agar food layer, which served as the interaction arena (*Figure 6—video 1; Figure*

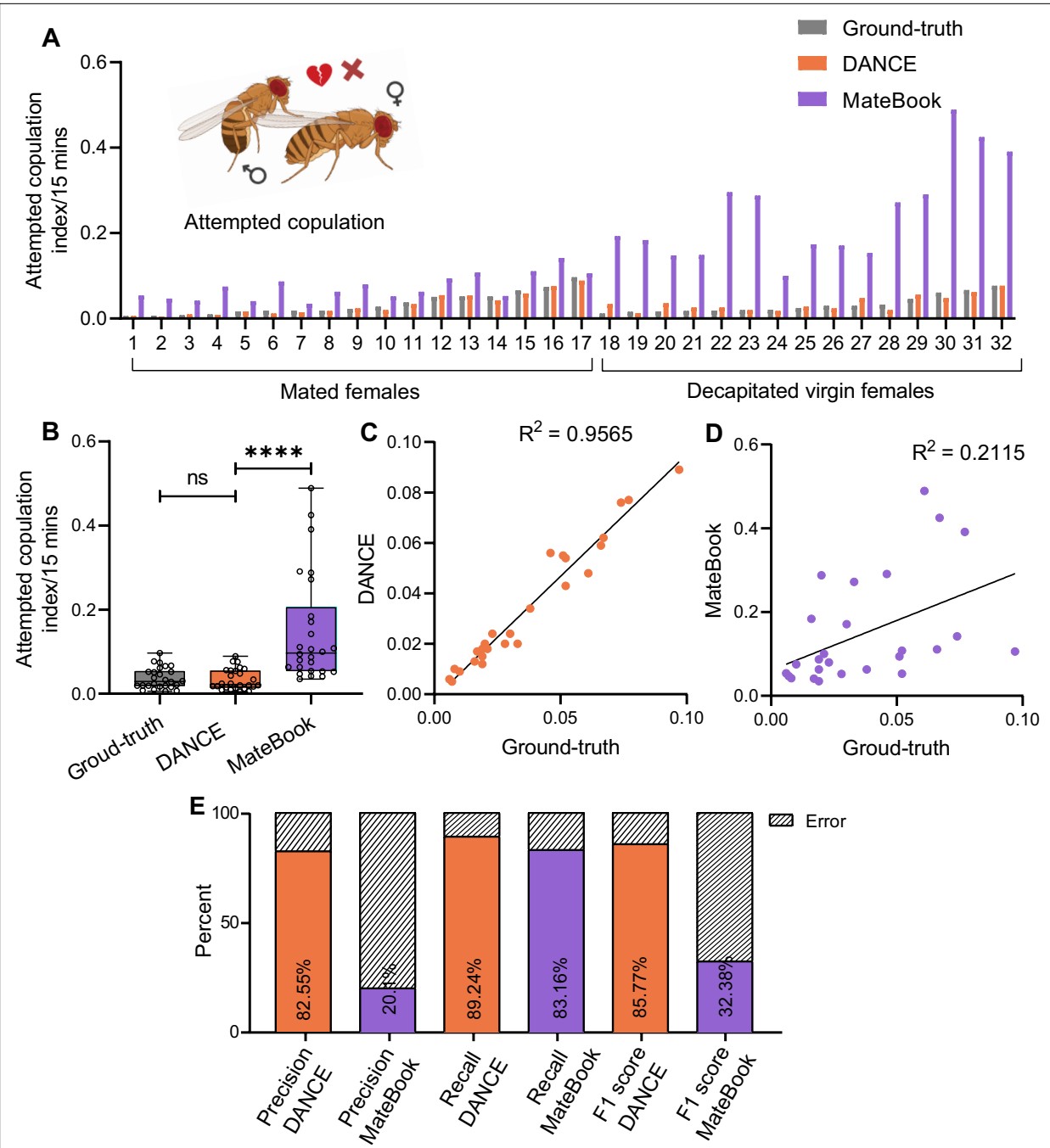

**Figure 4.** Validation of the Drosophila Aggression and Courtship Evaluator (DANCE) attempted-copulation classifier. (**A**) Attempted copulation index of males from 15-min-long videos scored using manual ground-truth annotations (gray), the 'DANCE attempted copulation classifier' (orange), and the MateBook (purple) with both mated and decapitated females (Friedman's ANOVA with Dunn's test: ground-truth vs. MateBook ****p<0.0001, ground-truth vs. DANCE ns, p>0.9999; n=32). (**B**) Comparison of ground-truth, DANCE attempted-copulation classifier, and MateBook scores (Kruskal–Wallis ANOVA with Dunn's test, ground-truth vs. DANCE ns, p>0.9999; ground-truth vs. MateBook ****p<0.0001, n=32). (**C**) Regression analysis of the attempted-copulation classifier vs. ground-truth ($R^2$=0.9565, n=32). (**D**) Regression analysis of MateBook vs. ground-truth ($R^2$=2115, n=32). (**E**) Precision, recall, and F1 score of the DANCE and MateBook attempted-copulation classifiers relative to the ground-truth scores.

The online version of this article includes the following source data and figure supplement(s) for figure 4:

**Source data 1.** Source data for *Figure 4* showing quantitative behavioral indices and performance metrics (bout-level analysis) for DANCE and existing methods used to quantify attempted-copulation behavior in both decapitated virgin and mated female flies.

**Figure supplement 1.** Evaluation of the Drosophila Aggression and Courtship Evaluator (DANCE) copulation classifier in the mixed-female dataset.

*Figure 4 continued on next page*

*Figure 4 continued*

**Figure supplement 1—source data 1.** Source data for *Figure 4—figure supplement 1* showing quantitative behavioral indices and performance metrics (bout-level analysis) for DANCE and existing methods used to quantify copulation behavior in mixed female dataset.

*6—video 2*). For courtship assays, the blister foil was bisected and fitted with a thin X-ray film separator comb that kept males and females apart until the start of recording, when the comb was removed to allow interaction (*Figure 6—video 3; Figure 6—video 4*). Because the tablet or smartphone screens used as the backlight generate heat, we placed a transparent acrylic spacer above the backlight to create a 4 mm air gap for heat dissipation (*Figure 6E–G*; *Figure 6—video 2; Figure 6—video 4*). This modification was essential for maintaining consistent behavioral recordings (*Figure 7*).

The DANCE hardware is intentionally modular, allowing users to adapt the setup to their experimental needs. The detailed assembly, cleaning, and reuse protocols are provided in the Materials and methods and on the GitHub page of the project.

## Benchmarking DANCE hardware

We benchmarked the DANCE hardware by applying the validated DANCE classifiers to videos recorded in the DANCE setup and comparing the results with those from established recording systems. Wild-type males displayed quantitatively similar levels of courtship and aggression in DANCE arenas and in pre-existing setups (*Figure 7*; *Figure 7—video 1*).

To test whether DANCE reproduces established behavioral findings, we examined the effects of social isolation and enrichment on aggression and courtship. Previous studies have shown that single-housing (SH) increases courtship attempts (*Dankert et al., 2009*; *Kim and Ehrman, 1998*; *Pan and Baker, 2014*) and promotes aggression (*Agrawal et al., 2020*; *Wang et al., 2008*; *Yadav et al., 2024*). We found that both the DANCE and pre-existing setups captured similar and statistically significant differences in courtship behaviors between SH and group-housed (GH) males (*Figure 7C–J*). These results confirm that DANCE reliably detects behavioral modulation by social experience.

We next compared an established aggression assay (*Dankert et al., 2009*; *Figure 7K*) with the DANCE aggression setup (*Figure 7L*). Both systems detected aggressive lunges in SH flies and showed consistent differences between SH and GH conditions (*Figure 7M–N*). Thus, DANCE hardware provides comparable sensitivity to conventional machine-vision-based setups while being more accessible.

We also tested whether diet composition alters aggression in DANCE assays, as nutrient availability and microbiome interactions can influence male aggression (*Jia et al., 2021*; *Lim et al., 2014*). Replacing yeast granules in the diet with yeast extract powder reduced baseline aggression (*Figure 7O*). These findings demonstrate that the DANCE hardware is sensitive enough to detect diet-dependent behavioral differences.

To test whether DANCE is compatible with neurogenetic manipulations, we used RNAi-mediated knockdown of the neuropeptide Drosulfakinin (Dsk) in insulin-producing neurons using the *dilp2*-GAL4 driver. Consistent with our previous findings (*Agrawal et al., 2020*), SH males with Dsk knockdown exhibited significantly increased aggressive lunges compared to controls (*Figure 7P and Q*).

We then evaluated DANCE's suitability for optogenetic assays, a common approach to dissect neural circuits underlying aggression (*Hoopfer et al., 2015*; *Wohl et al., 2023*; *Yadav et al., 2024*). An earlier study has shown that constitutive silencing of broad populations of dopaminergic neurons using *TH*-GAL4 produces unhealthy flies with impaired locomotion that rarely fight (*Alekseyenko et al., 2013*). We reasoned that transient, light-controlled silencing could overcome these limitations and tested this using an optogenetic module integrated with DANCE (*Figure 7—figure supplement 1A and B*; DANCE GitHub).

Optogenetic silencing of dopaminergic neurons expressing the green-light-sensitive anion channelrhodopsin GtACR1 (*Govorunova et al., 2015*; *Mohammad et al., 2017*) during 20 min interactions resulted in a significant increase in aggressive lunges in SH flies (*Figure 7R*). In addition, we observed higher frequencies of wing flicks and high-intensity aggressive behaviors such as boxing and tussling (*Figure 7—video 2*). Importantly, continuous silencing for 12 hr did not alter general locomotor activity between the GH and SH flies (*Figure 7—figure supplement 2*), confirming that these effects were not due to impaired movement.

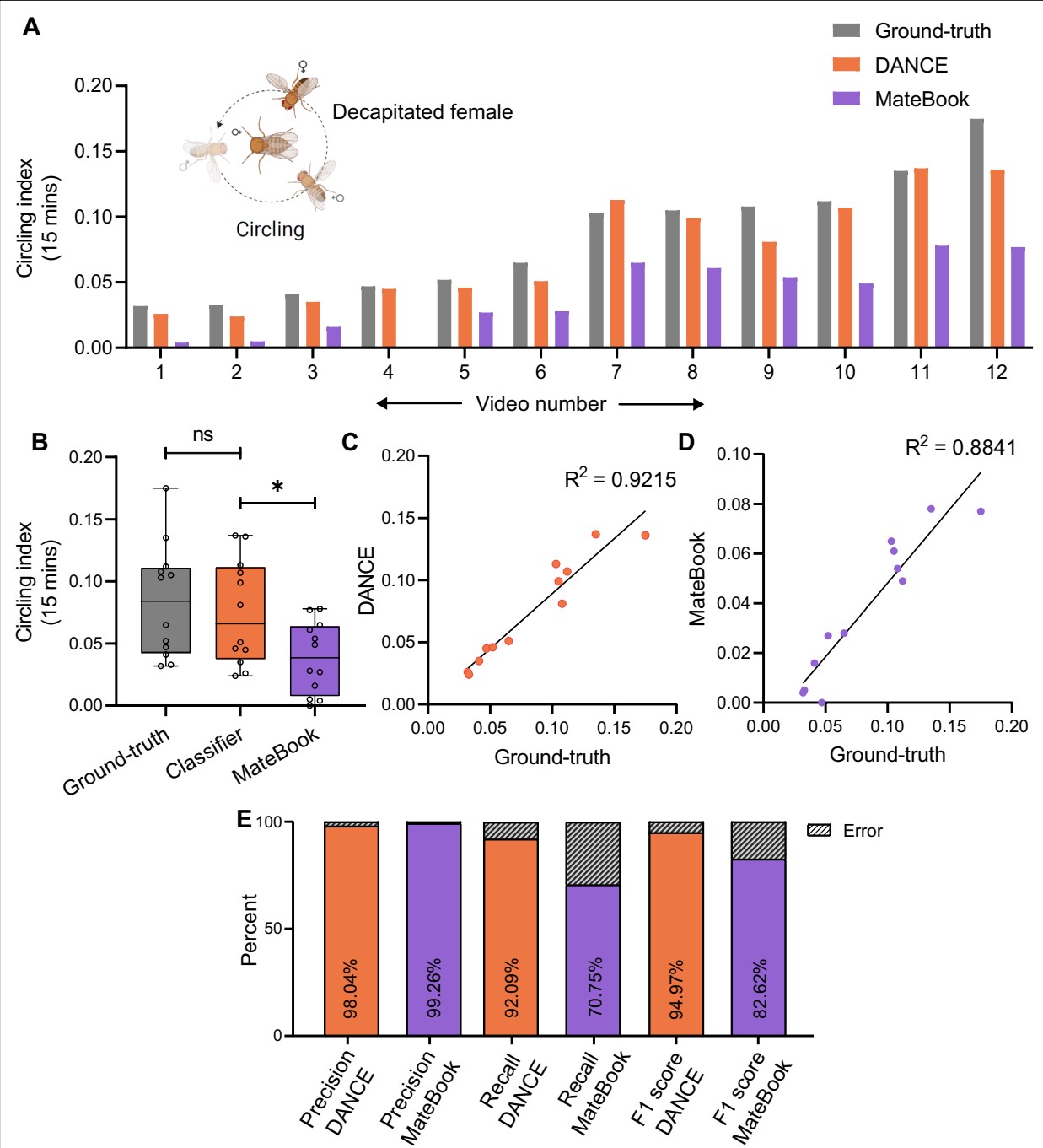

**Figure 5.** Evaluation of the <u>D</u>rosophila <u>A</u>ggression an<u>d</u> <u>C</u>ourtship <u>E</u>valuator (DANCE) circling classifier. (**A**) Circling index of males from 15-min-long videos scored using manual ground-truth annotations (gray), 'DANCE circling classifier' (orange), and MateBook (purple) with decapitated virgin females (Friedman's ANOVA with Dunn's test: ground-truth vs. DANCE ns, p=0.2049, ground-truth vs. MateBook ****p<0.0001, n=12). (**B**) Comparison of the ground-truth, DANCE, and MateBook circling classifiers (ordinary one-way ANOVA with Dunnett's test, ground-truth vs. DANCE ns, p=0.8014; ground-truth vs. MateBook *p=0.0157, n=12). (**C**) Regression analysis of the DANCE circling classifier vs. ground-truth ($R^2$=0.92, n=12). (**D**) Regression of MateBook vs. ground-truth ($R^2$=0.88, n=12). (**E**) Precision, recall, and F1 score of the DANCE and MateBook circling classifiers relative to the ground-truth score.

The online version of this article includes the following source data and figure supplement(s) for figure 5:

**Source data 1.** Source data for *Figure 5* showing quantitative behavioral indices and performance metrics (bout-level analysis) for DANCE and existing methods used to quantify circling behavior in decapitated virgin female flies.

**Figure supplement 1.** Evaluation of the <u>D</u>rosophila <u>A</u>ggression an<u>d</u> <u>C</u>ourtship <u>E</u>valuator (DANCE) circling classifier in the mated-female dataset.

*Figure 5 continued on next page*

*Figure 5 continued*

**Figure supplement 1—source data 1.** Source data for *Figure 5—figure supplement 1* showing quantitative behavioral indices and performance metrics (bout-level analysis) for DANCE and existing methods used to quantify circling behavior in mated female flies.

**Figure supplement 2.** Evaluation of the Drosophila Aggression and Courtship Evaluator (DANCE) following classifier in the mated-female dataset.

**Figure supplement 2—source data 1.** Source data for *Figure 5—figure supplement 2* showing quantitative behavioral indices and performance metrics (bout-level analysis) for DANCE and existing methods used to quantify following behavior in mated female flies.

**Figure supplement 3.** Frame-level analysis of duration-based courtship classifiers.

**Figure supplement 3—source data 1.** Source data for *Figure 5—figure supplement 3* showing performance metrics (frame-level analysis) for DANCE and existing methods used to quantify multiple courtship behaviors.

We also examined whether arena size influenced behavior. *TH*-GAL4>*UAS-GtACR1* males showed increased lunging vs. controls across all sizes tested (13, 17, and 21 mm; *Figure 7—figure supplement 3A–C*), indicating the optogenetic effect was robust to chamber dimensions. In courtship assays (11, 13, and 17 mm arenas; *Figure 7—figure supplement 4A–L*), single-housed males exhibited more

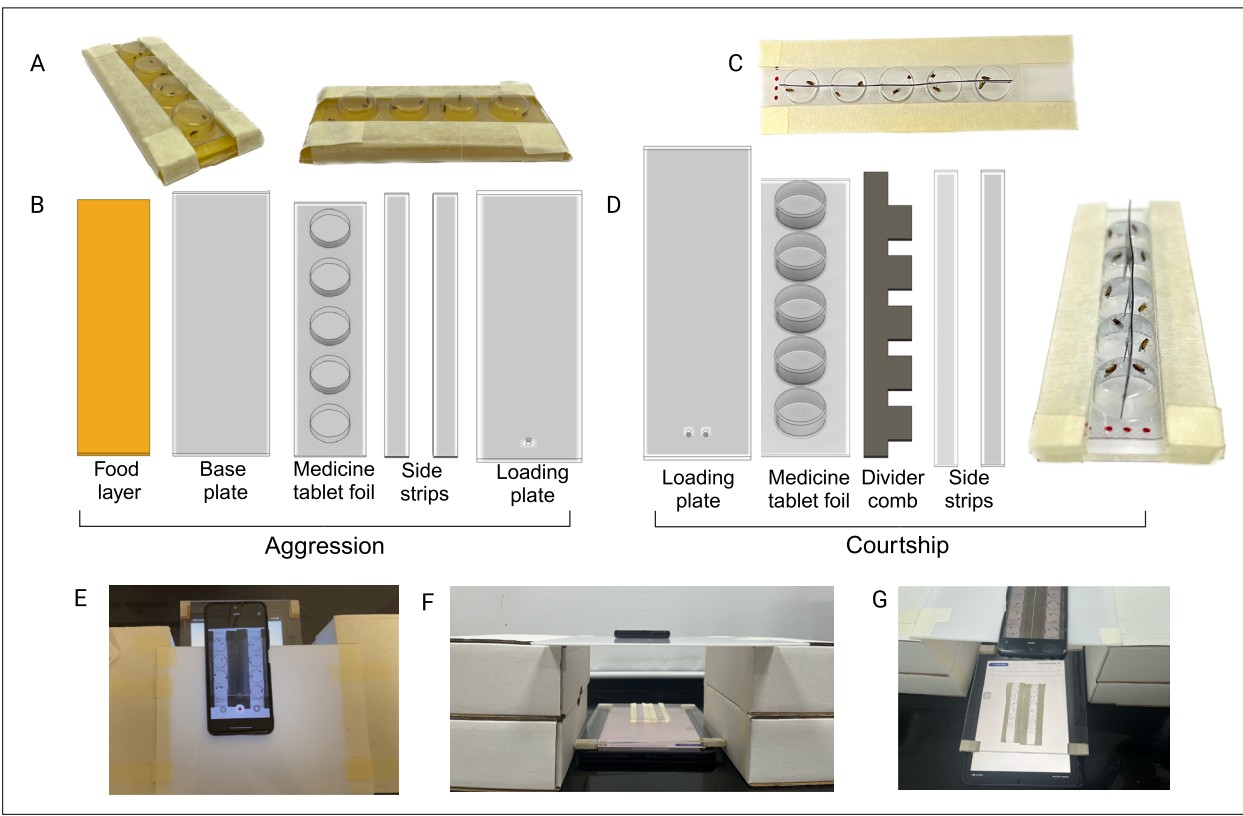

**Figure 6.** Drosophila Aggression and Courtship Evaluator (DANCE) hardware and recording setup. (**A**) DANCE aggression setup. (**B**) 3D-rendered components of the aggression setup. (**C**) DANCE courtship setup. (**D**) 3D-rendered components of the courtship setup, showing males and females separated by an X-ray film separator or 'divider comb'. (**E–G**) Top and side views of the DANCE setup with a smartphone camera for recording and an electronic tablet as the backlight. Created in BioRender.

The online version of this article includes the following video(s) for figure 6:

**Figure 6—video 1.** 3D-rendered Drosophila Aggression and Courtship Evaluator (DANCE) aggression hardware.
https://elifesciences.org/articles/105465/figures#fig6video1

**Figure 6—video 2.** Using the Drosophila Aggression and Courtship Evaluator (DANCE) hardware setup for recording aggression.
https://elifesciences.org/articles/105465/figures#fig6video2

**Figure 6—video 3.** 3D-rendered Drosophila Aggression and Courtship Evaluator (DANCE) courtship hardware.
https://elifesciences.org/articles/105465/figures#fig6video3

**Figure 6—video 4.** Using the Drosophila Aggression and Courtship Evaluator (DANCE) hardware setup for recording courtship.
https://elifesciences.org/articles/105465/figures#fig6video4

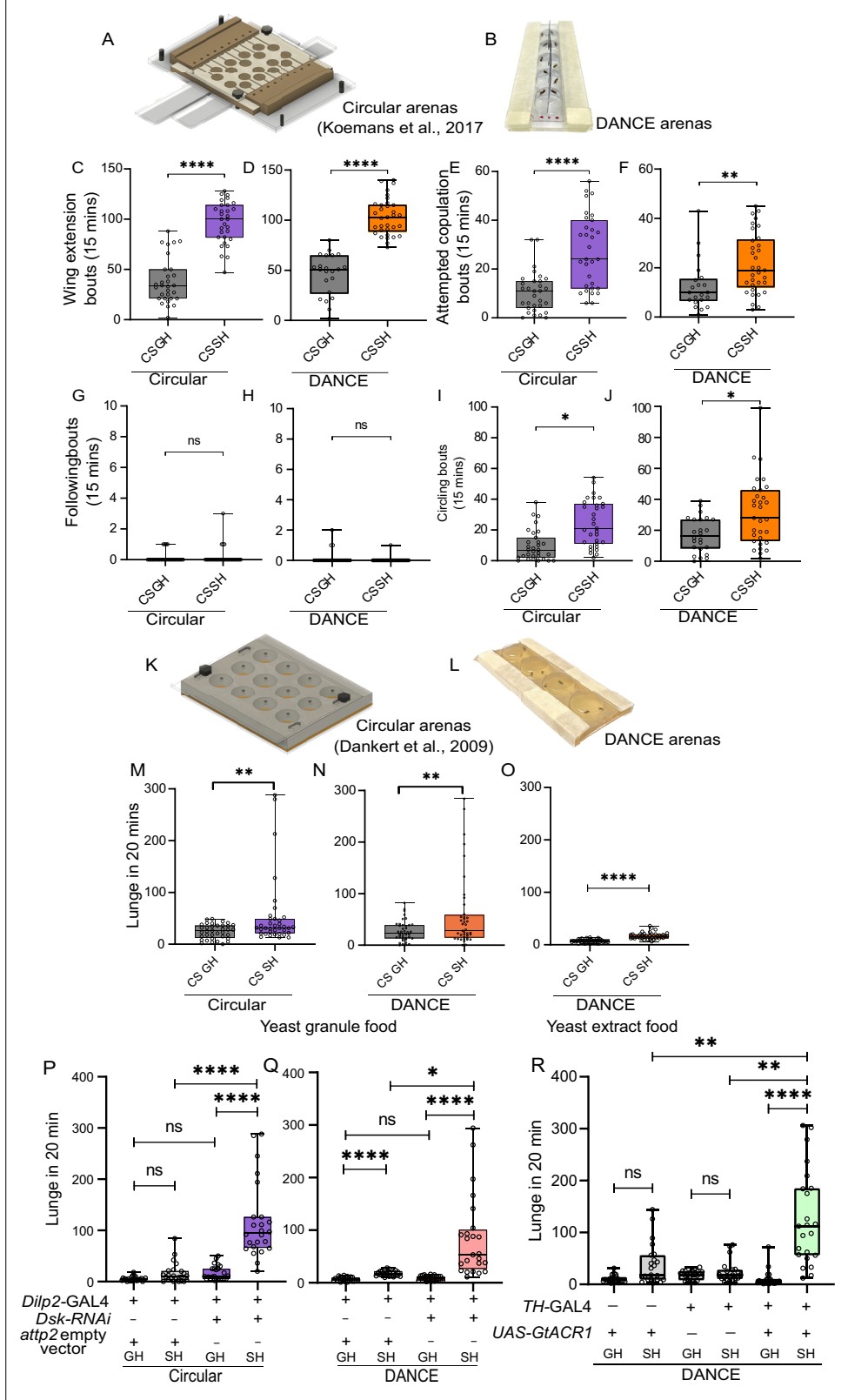

**Figure 7.** Benchmarking Drosophila Aggression and Courtship Evaluator (DANCE) hardware and application to neurogenetic tools. (**A–B**) Courtship behaviors recorded using a pre-existing circular setup (**Koemans et al., 2017**) and DANCE setup in group-housed (GH) and single-housed (SH) flies for (**C–D**) wing extension, (**C**) GH vs. SH ***p<0.0010, n=23; (**D**) GH vs. SH ****p<0.0001, GH, n=22 and SH, n=33. (**E–F**) Attempted copulation, (**E**) GH

*Figure 7 continued on next page*

*Figure 7 continued*

vs. SH ***p<0.0002, n=23; (**F**) GH vs. SH **p<0.0022, GH, n=21 and SH, n=33. (**G–H**) Following, (**G**) GH vs. SH ns, p>0.0959, n=23; (**H**) GH vs. SH ns, p<0.2537, GH, n=22 and SH, n=32. (**I–J**) Circling, (**I**) GH vs. SH *p<0.012, n=23; (**J**) GH vs. SH *p<0.0104, GH, n=24 and SH, n=31. (**K–L**) Aggressive lunges were recorded using a pre-existing circular setup (*Dankert et al., 2009*) and a DANCE setup. (**M–N**) Lunges of SH flies compared with those of GH flies reared on food with yeast granules. (**M**) GH vs. SH **p<0.0138, n=36; (**N**) GH vs. SH **p<0.0372, n=40. (**O**) Effect of yeast extract food on aggressive behavior; GH vs. SH ****p<0.0001, n=38–39. (**P–Q**) Genetic knockdown of the neuropeptide Drosulfakinin (Dsk) in insulin-producing neurons using *dilp2*-GAL4. (**P**) *Dilp2-GAL4*-GAL4>attp2 GH vs. SH ns, p<0.0502; *Dilp2-GAL4*-GAL4>attp2 GH vs. *Dilp2-GAL4*-GAL4>*Dsk* RNAi GH ns, p>0.9999; *Dilp2-GAL4*-GAL4>attp2 SH vs. *Dilp2-GAL4*-GAL4>*Dsk* RNAi SH ****p<0.0001; *Dilp2-GAL4*-GAL4>*Dsk* RNAi SH vs. *Dilp2-GAL4*-GAL4>*Dsk* RNAi SH ****p<0.0001; n=24. (**Q**) *Dilp2-GAL4*-GAL4>attp2 GH vs. SH ****p<0.0001, *Dilp2-GAL4*-GAL4>attp2 GH vs. *Dilp2-GAL4*-GAL4>*Dsk* RNAi GH ns, p>0.9999, *Dilp2-GAL4*-GAL4>*Dsk* RNAi SH vs. *Dilp2-GAL4*-GAL4>*Dsk* RNAi SH ****p<0.0001, *Dilp2-GAL4*-GAL4>attp2 SH vs. *Dilp2-GAL4*-GAL4>*Dsk* RNAi SH *p>0.0210, n=24. (**R**) Optogenetic silencing of dopaminergic neurons with *UAS-GtACR1* driven by the *TH*-GAL4 driver; *UAS-GtACR1* SH vs. GH ns, p=0.0986; *TH-GAL4* SH vs. GH ns, p=0.9999; *TH-GAL4>UAS-GtACR1* SH vs. GH ****p<0.0001; *UAS-GtACR1* SH vs. *TH-GAL4>UAS-GtACR1* SH **p<0.0012; *TH-GAL4* SH vs. *TH-GAL4>UAS-GtACR1* SH **p<0.0013; n=21–24. (C–J and M–O) Mann–Whitney U test; (P–R) Kruskal–Wallis test with Dunn's multiple comparisons.

The online version of this article includes the following video, source data, and figure supplement(s) for figure 7:

**Source data 1.** Source data for *Figure 7* showing for quantitative behavioral counts across aggression and courtship assays used to benchmark DANCE hardware and neurogenetic manipulations.

**Figure supplement 1.** Drosophila Aggression and Courtship Evaluator (DANCE) optogenetic recording setup for behavioral experiments.

**Figure supplement 2.** Effect of optogenetic silencing of dopaminergic neurons on daytime activity.

**Figure supplement 2—source data 1.** Source data for *Figure 7—figure supplement 2* showing daytime activity counts across days and housing conditions during optogenetic silencing of dopaminergic neurons.

**Figure supplement 3.** Effect of optogenetic silencing of dopaminergic neurons on male aggression across arena sizes.

**Figure supplement 3—source data 1.** Source data for *Figure 7—figure supplement 3* showing aggressive lunge counts during optogenetic silencing of dopaminergic neurons across different arena sizes.

**Figure supplement 4.** Quantification of wild-type *Drosophila* courtship behavior in Drosophila Aggression and Courtship Evaluator (DANCE) chambers of varying diameters.

**Figure supplement 4—source data 1.** Source data for *Figure 7—figure supplement 4* showing behavioral counts across multiple courtship behaviors in wild-type Drosophila measured in arenas of different sizes.

**Figure 7—video 1.** Aggression and courtship behaviors recorded in Drosophila Aggression and Courtship Evaluator (DANCE) hardware.
https://elifesciences.org/articles/105465/figures#fig7video1

**Figure 7—video 2.** Optogenetic silencing of dopaminergic neurons in Drosophila Aggression and Courtship Evaluator (DANCE) shows increased aggression.
https://elifesciences.org/articles/105465/figures#fig7video2

wing extension, attempted copulation, and circling than group-housed males, while following was similar. Although statistical tests were limited to within-size comparisons, the distributions suggest a modest decrease in interaction frequency in larger arenas, consistent with previous observations that larger chambers reduce encounter rates (*Chowdhury et al., 2021*).

Taken together, these results show that DANCE hardware provides reliable, reproducible behavioral measurements and is compatible with genetic and optogenetic manipulations, as well as environmental perturbations.

## Discussion

Here, we present the DANCE assay, an easy-to-use, modular, and robust analysis pipeline with inexpensive hardware to record and quantify aggression and courtship behaviors. We developed six novel behavioral classifiers using supervised machine learning to accurately quantify the aggression and courtship behaviors of *Drosophila* males. The hardware component of DANCE, fabricated from

repurposed and low-cost materials, provides a practical alternative to conventional machine-vision setups. The DANCE setup can be built for less than 0.30 USD, representing an approximately 10,000-fold reduction in cost compared with standard systems. Despite this simplicity, its performance was comparable to that of more specialized and expensive hardware, validating DANCE as a reliable and accessible behavioral platform. This accessibility enables rapid behavioral screening and wider adoption by the neuroscience community, including resource-limited laboratories and teaching environments.

Various components of the DANCE assay, such as the behavioral classifiers, hardware design, and analysis code, are publicly available and can be used independently. This open, modular design gives researchers the flexibility to customize classifiers for specific behavioral paradigms or incorporate new data without developing a classifier from scratch. Although not implemented here, DANCE can be readily extended to real-time feedback experiments using open-source reactive programming tools such as Bonsai (*Lopes et al., 2015*). This framework can also support the development of future classifiers for additional social behaviors, including aggressive acts such as fencing, wing flicking, tussling, chasing, or female headbutting (*Chen et al., 2002*; *Nilsen et al., 2004*), and courtship behaviors such as male tapping, licking, or female rejection (*Sokolowski, 2001*).

Such high-resolution analysis of complex social interactions can provide deeper understanding of mating dynamics, sexual selection, and the influence of genetics and evolution. Further, quantifying behavioral dynamics can reveal the temporal organization of individual components of behavior (*Nilsen et al., 2004*; *Seeds et al., 2014*; *Simon and Heberlein, 2020*; *Zhang et al., 2020*). DANCE can also serve as a flexible framework for studying complex behaviors across multiple *Drosophila* species and other insects, enabling comparative and evolutionary analyses. The adaptability and portability of the DANCE assay make it particularly useful for ethologists examining insect behavior in seminatural or field-like environments. Together, these features position DANCE as a bridge between laboratory-based ethology and ecological studies of natural behavior.

# Materials and methods

**Key resources table**

| Reagent type (species) or resource | Designation | Source or reference | Identifiers | Additional information |
|---|---|---|---|---|
| Chemical compound, drug | Sigmacote | Sigma-Aldrich | Cat#: SL2 | |
| Chemical compound, drug | Alcojet | Alconox | Cat#: 1401-1 | |
| Chemical compound, drug | Sucrose | HiMedia | Cat#: GRM601 | |
| Chemical compound, drug | Agar | HiMedia | Cat#: GRM026 | |
| Chemical compound, drug | Yeast extract powder | HiMedia | Cat#: RM0271 | |
| Chemical compound, drug | Yeast granules | AB Mauri, India | | |
| Chemical compound, drug | Apple juice | Commercial | | |
| Chemical compound, drug | Fluon (Insect-a-Slip) | BioQuip | Cat#: 2871B | |
| Strain, strain background (*Drosophila melanogaster*, male) | Canton-S | Ulrike Heberlein (HHMI Janelia) | | Wild-type strain |
| Genetic reagent (*Drosophila melanogaster*) | TH-GAL4 | Bloomington Drosophila Stock Center | RRID:BDSC_51982 | |
| Genetic reagent (*Drosophila melanogaster*) | dilp2-GAL4 | Bloomington Drosophila Stock Center | RRID:BDSC_37516 | |
| Genetic reagent (*Drosophila melanogaster*) | Dsk-RNAi | Bloomington Drosophila Stock Center | RRID:BDSC_25869 | |
| Genetic reagent (*Drosophila melanogaster*) | attP2 empty vector control | Bloomington Drosophila Stock Center | RRID:BDSC_36303 | |
| Genetic reagent (*Drosophila melanogaster*) | UAS-GtACR1 | *Mohammad et al., 2017* | RRID:BDSC_92983 | Gift from Gaurav Das, (NCCS, Pune) |

*Continued on next page*

*Continued*

| Reagent type (species) or resource | Designation | Source or reference | Identifiers | Additional information |
|---|---|---|---|---|
| Software, algorithm | JAABA | *Kabra et al., 2013* | RRID:SCR_027430 | https://jaaba.sourceforge.net/ |
| Software, algorithm | Caltech FlyTracker | *Eyjolfsdottir et al., 2014* | RRID:SCR_027431 | https://kristinbranson.github.io/FlyTracker/ |
| Software, algorithm | MateBook | *Ribeiro et al., 2018* | | https://github.com/Dicksonlab/MateBook |
| Software, algorithm | CADABRA | *Dankert et al., 2009* | | https://www.vision.caltech.edu/cadabra/ |
| Software, algorithm | GraphPad Prism 8 | GraphPad Software | RRID:SCR_002798 | http://www.graphpad.com/ |
| Software, algorithm | DANCE classifiers and code | This paper | RRID:SCR_027812 | https://github.com/agrawallab/DANCE |
| Software, algorithm | BioRender | BioRender Software | RRID:SCR_018361 | https://www.biorender.com/ |
| Software, algorithm | Inkscape | Inkscape Software | RRID:SCR_014479 | https://github.com/inkscape/inkscape |
| Other | LEDs (520–540 nm) | Lumileds, SM Electronic Technologies Pvt Ltd. Bangalore | Model: 2835 | Hardware and equipment |
| Other | DMK 33UX252 USB 3.0 monochrome camera | Menzel Vision and Robotics Pvt Ltd. Mumbai | Model: DMK 33UX252 | Hardware and equipment |
| Other | Metaphase backlight | Alpha Techsys, Pune | Model: TMS, BHS4-00100-X-W-24V | Hardware and equipment |
| Other | Huawei Y9 2019 smartphone | Huawei | Model: Y9 2019 | Hardware and equipment |
| Other | OnePlus Nord CE 2 Lite 5G smartphone | OnePlus | Model: CPH2381 | Hardware and equipment |
| Other | Redmi Note 11 Pro+ 5G smartphone | Xiaomi | Model: 221116SI | Hardware and equipment |
| Other | iPad Air (5th Generation) | Apple | Model: iPad Air 5 | Hardware and equipment |
| Other | iPhone 13 | Apple | Model: 13 | Hardware and equipment |
| Other | White-screen light app | App Store/Play Store | | Hardware and equipment |

The details of all the custom codes, analysis pipelines, sample files used to run the analysis and DANCE classifiers are available in our GitHub repository at https://github.com/agrawallab/DANCE, copy archived at *Dey and Agrawal, 2025*.

## Fly husbandry

Flies were reared on standard food at 25°C and 65% relative humidity with a 12 hr:12 hr light–dark cycle. All the assays were performed at 25°C with 65% relative humidity, unless mentioned otherwise. For the aggression and courtship experiments, Canton-S (CS) male flies were collected within 24 hr of eclosion and housed in groups (20 male flies per vial, 90 mm in length and 25 mm in diameter) or isolated (1 male fly per vial, 70 mm in length and 10 mm in diameter) for 6 days. Randomization was not performed for behavioral experiments because flies were grouped by genotype and housing condition, and all cohorts were handled identically.

The following fly lines were acquired from the Bloomington Drosophila Stock Center (BDSC), USA: *TH*-GAL4 (RRID:BDSC_51982), *Dilp2*-GAL4 (RRID:BDSC_37516), *Dsk*-RNAi (RRID:BDSC_25869), and attP2 empty vector control (RRID:BDSC_36303). *UAS-GtACR1* flies were a gift from Gaurav Das, NCCS, Pune, India, and CS flies were obtained from Ulrike Heberlein, HHMI, Janelia Research Campus, Ashburn, VA, USA.

## Aggression assay

Aggression assays were performed as described previously (*Dankert et al., 2009*; *Dierick, 2007*). In brief, the behavioral chamber is made up of 12 well-aggressive arenas (10 mm in height and 16 mm in diameter/arena). These arenas were covered by a sliding lid with 2 mm loading holes to facilitate the introduction of flies. A pair of male flies that were housed either as GH or SH was introduced into the arena wells by gentle mouth aspiration through the loading holes. After the flies were loaded, the sliding lid was tightened with screws. Fluon (Insect-a-slip, Bioquip: Cat#: 2871B) was applied to the arena walls, which were left to dry overnight to create a slippery surface and prevent climbing. Sigmacote (Sigma-Aldrich: SL2) was used to coat the sliding lid to reduce walking on the arena ceiling. The chamber was placed on a food plate containing commercial apple juice (without added sugars), 2.5% wt/vol sucrose (HiMedia: GRM601), and 2.25% wt/vol agar (HiMedia: GRM026). For experiments to test the effects of fly food nutrients, either 2.4% yeast extract powder (HiMedia: RM0271) or 2.4% yeast granules (Prime Instant Dry Yeast, AB Maury, India) were mixed in the fly food. Optogenetic experiments were performed essentially as described earlier (*Govorunova et al., 2015*; *Mohammad et al., 2017*). LEDs emitting 520–540 nm light, peak emission 530 nm (Lumileds: High Power LEDs— single color: L128-GRN1003500000) were controlled via an Arduino microcontroller and powered through a computer to deliver illumination onto the blister-pack arenas. The LEDs were used at an intensity of 0.0004 μW, which was measured with a power meter (Newport: 843R). For all the aggression assays, the flies were allowed to acclimatize in the arena for 5 min, after which the activity was recorded for 20 min. The assays were performed during ZT0–ZT2.5, i.e., during the first 2.5 hr of the morning activity peak.

## Courtship assay

Courtship assays were performed as described previously (*Koemans et al., 2017*; *Ribeiro et al., 2018*). Single pairs of males and females were introduced into individual arenas of an 18-well courtship chamber (10 mm diameter). Male and female flies were introduced into one half of the chamber by sliding entry holes with a removable separator that divided the chamber into two halves. The flies were allowed to acclimatize to the arena for 5 min, after which the separator was removed, and courtship behavior was recorded for 15 min at 30 fps using a white backlight. The assay was performed from ZT0 to ZT3 or ZT9 to ZT12 (during peak activity windows). For mated females, 20 females were housed with 10 males for 4–6 days. For decapitated virgin females, 2- to 4-day-old virgins were anesthetized with $CO_2$ and decapitated immediately before the assay.

## DANCE hardware

Circular transparent medicine blister packs serve as aggression or courtship arenas. Blister packs were mounted on 2 mm acrylic base plates and secured with paper tape. The arena dimensions were as follows: aggression, 13 mm × 5.5 mm (diameter ×height) (also tested: 17 mm×4 mm and 21 mm×8 mm); courtship, 11 mm × 4.5 mm (also tested: 13 mm×5.5 mm and 17 mm×4 mm). Arena walls and roofs were coated with Sigmacote (Sigma-Aldrich: SL2) using cotton swabs (Solimo, Amazon India) to prevent flies from climbing.

A thin 2 mm acrylic base plate carrying the food layer (2.25% wt/vol agar in commercial apple juice with 2.5% wt/vol sucrose) and a side spacer were assembled and held together with paper tape (*Figure 6—video 1*). The tip of the food plate was covered with paper tape to allow smooth sliding of the blister foil without damaging the food surface. Flies were introduced through loading holes (2 mm diameter), and paper tape strips were used to prevent food damage during assembly and to seal small gaps to avoid escape (*Figure 6—video 2*).

Arenas were reusable for approximately 30–40 times when washed in 0.05% Alcojet (Alconox: 1401-1) and air-dried. Heat exposure during cleaning was avoided to prevent deformation.

The standard DANCE courtship arena used transparent medicine blister packs measuring 11 mm in diameter and 4.5 mm in height. Additional arena sizes of 13 mm×5.5 mm and 17 mm×4 mm (diameter × height) were also tested. The top surface of each arena strip (five wells in a row) was modified by cutting thin slits with a sharp razor to insert a separator comb made from repurposed X-ray film. The foil and 2 mm acrylic base plate, which contained a loading hole (2 mm diameter), were joined with paper tape. The courtship base plate contained two loading holes so that male and female flies could

be introduced on either side of the separator comb. The assembly was sealed at both ends with paper tape to prevent gaps from escaping (*Figure 6—video 3*).

Before recording, the separator comb was gently lifted to allow interaction between the flies, taking care to keep the slit edges intact and avoid distortion during filming (*Figure 6—video 4*). Blister packs modified with slits and separator combs are typically reusable for ~10–15 experiments, after which the slits tend to widen, and new blister packs are recommended.

## Video acquisition

For the traditional setup, the interaction of the flies was recorded using machine-vision cameras (DMK 33UX252 USB 3.0 monochrome camera). White backlight (TMS, BHS4-00-100-X-W-24V) provided the light source for both the courtship and aggression experiments. Videos were recorded at 30 frames per second (fps) for 15 min for courtship or 20 min for aggression in H.264 (.mp4) format with 1440×1080 resolution. These videos were used for training, testing, and validating the DANCE classifiers.

For DANCE hardware testing, various Android smartphone cameras were used (Huawei Y9 2019; OnePlus Nord CE 2 Lite 5G, model: CPH2381; Redmi Note 11 Pro+ 5G, model: 221116SI) at 30 fps, 1080p resolution in H.264 (.mp4) format for 15 min (courtship) or 20 min (aggression). An electronic tablet (iPad Air, 5th Generation) or smartphone (iPhone 13) running a 'white screen light app' served as the background illumination source. The screen brightness was adjusted within the app to optimize contrast for fly tracking. A transparent acrylic sheet with 4 mm spacers was kept on top of this 'backlight' to create an air gap to ensure heat exchange and prevent the DANCE arenas from becoming hot. Devices were recommended to be placed in airplane mode during recordings to avoid interruptions.

## Tracking flies using FlyTracker

Fly locations, body orientations, and interactions were tracked using Caltech FlyTracker (*Eyjolfsdottir et al., 2014*). These data were then pushed to the JAABA pipeline to develop DANCE classifiers. Occurrences of identity switches were corrected using the FlyTracker 'visualizer' identity-correction tool. Tracking accuracy and identity swap quantification were validated by a semi-manual inspection of flagged frames and intervals (see *Supplementary file 5* for details).

## Pre-existing algorithms used for benchmarking

### CADABRA

CADABRA (*Dankert et al., 2009*) analyzes two-fly interactions based on spatial and postural features, classifying lunges, wing threats, circling, wing extension, and copulation based on fixed rules. The specific CADABRA definitions are described in the subsequent section.

### Divider assay

The Divider assay (*Chowdhury et al., 2021*) uses a 3D-printed rectangular chamber with 12 arenas (13 mm × 4.5 mm, W×H), each separated by an opaque divider. Behavioral data are analyzed with a custom FlyTracker-JAABA pipeline. Since the Divider assay classifier was trained on recordings from a rectangular geometry, it can affect transferability to circular arenas.

### MateBook

MateBook (*Ribeiro et al., 2018*; https://github.com/Dicksonlab/MateBook copy archived at *Arthur, 2025*) uses machine vision to track flies and classify male courtship behaviors (following, wing extension, orientation, copulation, and circling). The outputs include a .tsv file with bout statistics and an ethogram. For comparison with the DANCE classifiers, the MateBook persistence filters were adjusted so that the minimum bout duration threshold was 0.33 s (10 frames at 30 fps) for all behaviors except copulation, which retained a 45 s threshold. This adjustment ensured comparable persistence criteria between the MateBook and DANCE analyses.

### Developing DANCE classifiers

JAABA (*Kabra et al., 2013*) was used to train classifiers iteratively: true bouts were labeled, obvious non-bouts assigned as 'None', and false positives were relabeled until performance plateaued.

The DANCE lunge classifier was trained on 11 independent videos with classifier accuracy improving progressively from Video 1 to Video 9. Peak performance was achieved after inclusion of the ninth training video, which provided the best balance of precision, recall, and F1 score. Adding further data (Videos 10 and 11) did not enhance classifier accuracy and instead produced a slight reduction in precision and recall, likely due to increased behavioral variability across sessions. Therefore, the final lunge classifier was trained on nine videos, which yielded the most robust and generalizable model.

Courtship classifiers were developed following the same iterative procedure. Training sets included videos with both decapitated and intact females (mated or virgin) to capture behavioral variability and ensure robust generalization. Independent classifiers were trained for wing extension, following, circling, attempted copulation, and copulation. All classifiers were validated using manually annotated 'ground-truth' test videos that were not included in the training set.

Courtship training sets included decapitated virgin videos to enrich attempted copulation and circling bouts and to balance 'None' class examples for following and copulation. Wing extension used mated females only.

The training data volumes (frames and approximate durations) were as follows:

| Behavior | Frames | Approx. duration (s) |
| --- | --- | --- |
| Lunge | 1449 | ~48 |
| Wing extension | 99,947 | ~3332 |
| Attempted copulation | 39,513 | ~1317 |
| Copulation | 56,979 | ~1899 |
| Circling | 14,396 | ~480 |
| Following | 25,787 | ~860 |

Test videos were manually annotated in JAABA ground-truthing mode before any classifier predictions were examined, and annotations were performed independently of classifier outputs.

## Manual behavioral annotations and inter-annotator reliability

Manual annotations ('ground-truth') were generated using JAABA's ground-truthing mode by labeling behavioral bouts frame by frame. To assess observer bias, subsets of videos were annotated independently by two evaluators and compared using non-parametric tests. Where relevant, the results describe inter-annotator comparisons. Blinding was not performed because genotypes and experimental conditions were known during experiments and analysis.

## Characterization of male aggression and courtship behaviors

The DANCE classifiers were trained and validated in JAABA using established behavioral definitions from previous studies (*Dankert et al., 2009*; *Ribeiro et al., 2018*), with input from experienced users (*Supplementary file 3*).

### Behavior definitions

#### Lunge
As defined by *Dankert et al., 2009*; *Ribeiro et al., 2018*, 'the attacking fly rises on hind legs, lifting its long body axis by 45°, then snaps down on its opponent's body with its head at ~200 mm/s'.

#### Wing extension
The angle between the body axis and wing tip line exceeds 30°, persisting for at least 13 frames (0.5 s at 25 fps) (*Dankert et al., 2009*; *Ribeiro et al., 2018*).

#### Attempted copulation
Abdominal curling without mounting, or mounting lasting ≥0.33 s but <45 s (10–1350 frames at 30 fps).

## Copulation

Mounting lasting ≥45 s (>1350 frames at 30 fps), adapted from *Dankert et al., 2009*; *Ribeiro et al., 2018*.

## Circling

Sideways drift around the female in a circular path at constant velocity, persisting for ≥13 frames (*Dankert et al., 2009*; *Ribeiro et al., 2018*).

## Following

The male remains 2–5 mm behind the female while both walk at ≥2 mm/s, persisting for ≥25 frames (*Dankert et al., 2009*; *Ribeiro et al., 2018*). To account for variability in female mating condition and body size, training datasets included videos of males paired with decapitated virgin, intact virgin, and mated females. For the copulation classifier, videos in which males were paired with virgin females to capture prolonged occlusion events characteristic of mating were used for training.

For duration-based classifiers, a post-processing filter was applied to exclude bouts shorter than 98% of those observed in manual annotations, ensuring consistency with human-defined behavioral durations.

This framework can be readily adapted by the research community to develop additional behavioral classifiers. Owing to file size limitations, training videos are not hosted online but are available upon request.

## Manual behavioral annotations

To quantitatively evaluate classifier performance, manual behavioral annotations ('ground-truth') were generated using JAABA's ground-truthing mode (*Kabra et al., 2013*). Classifier robustness was assessed on unseen videos comprising the testing set. Each testing video was first manually annotated by identifying behavioral bouts as 'true behavior', independent of the classifier's output. These same videos were then processed through the trained classifier using JAABAPlot, and the results were compared as described below.

## Comparison of manual and DANCE annotations

The classifier outputs were compared with manually annotated ground-truth data at both the bout and frame levels, depending on the classifier type. For the single-frame lunge classifier, comparisons were based on total bout counts, whereas for all duration-based courtship classifiers (wing extension, following, circling, attempted copulation, and copulation), comparisons were made using bout durations and frame-level annotations.

For each assay, a behavioral index was calculated as the proportion of frames in which the male engaged in the specified behavior. This was obtained by dividing the total number of frames annotated for that behavior by the total number of frames in the recording. Regression analyses and performance metrics were computed using either bout counts or behavioral indices, depending on the classifier type, in GraphPad Prism 8 (GraphPad Software). Manual annotations and classifier outputs were compared to identify true positives (TP), false positives (FP), and false negatives (FN), at either frame level or bout level.

## Bout-level analysis

A predicted bout was scored as a TP if it overlapped with a ground-truth bout by at least one frame (~33 ms at 30 fps), consistent with previous studies (*Leng et al., 2020*). When multiple predicted bouts overlapped a single ground-truth bout, they were collectively counted as one TP. Conversely, when a single predicted bout overlapped multiple ground-truth bouts, the TP count equaled the number of ground-truth bouts. Predicted bouts with no overlap were scored as FP, and ground-truth bouts with no overlapping prediction were scored as FN. These same criteria were applied to both bout-level and frame-level evaluations, with the latter accounting for the total number of frames contributing to TP, FP, and FN classifications to provide a more granular measure of accuracy (see *Figure 5—figure supplement 3*).

### Frame-level analysis

A predicted frame was scored TP if it matched the ground-truth frame; frames predicted as behavior only by the classifier were FP; frames annotated as behavior only by ground-truth were FN. These frame-level TP, FP, FN were then similarly used to calculate precision, recall, and F1 score, providing a more granular measure of classifier accuracy.

### Performance metrics

A custom Python script was used to calculate overlaps and derive standard classification metrics. The precision, recall, and F1 scores were computed using the following formulas:

$$precision = \frac{(number\ of\ true\ positives)}{(number\ of\ true\ positives) + (number\ of\ false\ positives)}$$

$$recall = \frac{(number\ of\ true\ positives)}{(number\ of\ true\ positives) + (number\ of\ false\ negatives)}$$

$$F1\ score = 2\ x\ \frac{(precision\ x\ recall)}{(precision + recall)}$$

Precision represents the fraction of correctly predicted positive observations among all predicted positives, recall represents the fraction of correctly predicted positive observations among all actual positives, and the F1 score provides the harmonic mean of precision and recall. These metrics were used to quantify and compare the performance of the DANCE classifiers against manually annotated ground-truth and existing algorithms.

For duration-based behaviors, the behavioral index was used as a continuous variable in regression and performance analyses. For the lunge classifier, which identifies discrete one-frame events, comparisons were made using total bout counts.

### Statistical analysis

All the statistical analyses were performed using *GraphPad Prism 8*, custom Python scripts, or *Microsoft Excel*. For non-normally distributed data, non-parametric tests such as the Mann–Whitney U test or Kruskal–Wallis ANOVA with appropriate post hoc corrections were used. Formal power calculations were not performed, as sample sizes were chosen based on established standards in *Drosophila* behavioral studies. No animals or data points were excluded from analysis due to attrition; all recorded flies were included unless excluded a priori based on predefined criteria (e.g. physical injury or tracking failure).

## Acknowledgements

We thank Barry Dickson (Queensland Brain Institute, Australia) for insightful discussions and suggestions related to MateBook and Ben Arthur (HHMI, Janelia Research Campus, USA) for guidance in setting up the MateBook analysis. We are grateful to Mayank Kabra, Kristin Branson, and colleagues for developing JAABA and for their ongoing support to the user community. We thank Ulrike Heberlein (HHMI, Janelia Research Campus, USA), Gaurav Das (NCCS, Pune), and the Bloomington Drosophila Stock Center (NIH P40OD018537) for providing fly stocks. We acknowledge Santhosh Chidangil (MAHE) for assisting with the LED power measurements and Santosh D'Mello (LSU Shreveport) for helpful discussions. We are also grateful to Gaurav Das and Toshiharu Ichinose (Tohoku University, Japan) for critical reading and feedback on the manuscript. Figures were created with BioRender.com and Inkscape. This work was supported by funding to PA from the Department of Biotechnology (DBT), Ramalingaswami Re-entry Fellowship (BT/RLF, Re-entry/34/2018), and DBT, Research grant (BT/PR36166/BRB/10/1859/2020) by the DBT, Ministry of Science and Technology, Government of India. RSPY was supported by a TMA Pai fellowship from MAHE and DBT, Research grant to PA. FA was supported by Ramalingaswami fellowship, DBT, India to PA. TK was supported by DBT, Research grant to PA. MV is supported by NFST, Ministry of Tribal Affairs, Government of India. SA is supported by the Department of Science and Technology (DST) INSPIRE Fellowship. PPP is supported by a TMA Pai fellowship from MAHE, India. SBS is supported by the Anusandhan National Research Foundation (ANRF), Ministry of Science and Technology, Government of India, grant to PA (CRG/2022/006846).

# Additional information

## Competing interests

R Sai Prathap Yadav, Faizah Ansari, Pavan Agrawal: RSPY, FA and PA are listed as inventors on a published patent related to this work by the Indian patent office, titled 'Device For Measuring Complex Social Behaviors In Small Insects', Application No. 202441072884. The other authors declare that no competing interests exist.

## Funding

| Funder | Grant reference number | Author |
|---|---|---|
| Department of Biotechnology, India | BT/RLF | Pavan Agrawal Faizah Ansari |
| Department of Biotechnology, India | Re-entry/34/2018 | Pavan Agrawal Faizah Ansari |
| Department of Biotechnology, India | BT/PR36166/ BRB/10/1859/2020 | Pavan Agrawal R Sai Prathap Yadav Tanvi Kottat |
| Anusandhan National Research Foundation (ANRF), Ministry of Science and Technology, Government of India | CRG/2022/006846 | Pavan Agrawal |
| NFST, Ministry of Tribal Affairs, Government of India | 201516-NFST-2015-17-ST-TEL-806-RENEWAL-2021-22 | Manohar Vasam |
| MAHE, India | TMA Pai fellowship | P Pallavi Prabhu R Sai Prathap Yadav |
| Department of Science and Technology (DST) | DST/ INSPIRE/03/2023/001786 | Shrinivas Ayyangar |

The funders had no role in study design, data collection and interpretation, or the decision to submit the work for publication.

## Author contributions

R Sai Prathap Yadav, Paulami Dey, Data curation, Software, Formal analysis, Validation, Investigation, Visualization, Methodology, Writing – original draft, Writing – review and editing; Faizah Ansari, Data curation, Software, Formal analysis, Validation, Visualization, Methodology, Writing – original draft, Writing – review and editing; Tanvi Kottat, Data curation, Software, Formal analysis, Validation, Visualization, Writing – review and editing; Manohar Vasam, Validation, Visualization, Writing – review and editing; P Pallavi Prabhu, Shrinivas Ayyangar, Swathi Bhaskar S, Validation, Writing – review and editing; Krishnananda Prabhu, Writing – review and editing; Monalisa Ghosh, Investigation, Writing – review and editing; Pavan Agrawal, Conceptualization, Resources, Supervision, Funding acquisition, Visualization, Methodology, Writing – original draft, Project administration, Writing – review and editing

## Author ORCIDs

R Sai Prathap Yadav ⓘ http://orcid.org/0009-0005-1577-1945
Paulami Dey ⓘ https://orcid.org/0000-0003-4303-4510
Faizah Ansari ⓘ https://orcid.org/0000-0002-6453-7179
Manohar Vasam ⓘ https://orcid.org/0000-0002-3148-8729
Krishnananda Prabhu ⓘ https://orcid.org/0000-0003-3479-9597
Pavan Agrawal ⓘ https://orcid.org/0000-0002-4494-7859

Reviewer #1 (Public review): https://doi.org/10.7554/eLife.105465.3.sa1
Reviewer #2 (Public review): https://doi.org/10.7554/eLife.105465.3.sa2
Reviewer #3 (Public review): https://doi.org/10.7554/eLife.105465.3.sa3

Author response https://doi.org/10.7554/eLife.105465.3.sa4

## Additional files

### Supplementary files

Supplementary file 1. Bill of materials for the _Drosophila_ Aggression and Courtship Evaluator (DANCE) setup and comparison with existing setups.

Supplementary file 2. Comparison between the _Drosophila_ Aggression and Courtship Evaluator (DANCE) lunge classifier, ground-truth, and existing methods.

Supplementary file 3. Definitions of the behavioral classifiers.

Supplementary file 4. Comparison between the _Drosophila_ Aggression and Courtship Evaluator (DANCE) courtship classifiers, ground-truth, and MateBook.

Supplementary file 5. Quantifying identity swaps to validate tracking across setups.

MDAR checklist

### Data availability

Source data files have been provided for all quantitative analyses presented in the figures. Figure 2-source data 1, Figure 2-figure supplement 1-source data 1, and Figure 2-figure supplement 2-source data 1 contain the numerical data used to generate Figure 2 and its supplements. Figure 3-source data 1 and Figure 3-figure supplement 1-source data 1 contain the numerical data used to generate Figure 3 and its supplement. Figure 4-source data 1 and Figure 4-figure supplement 1-source data 1 contain the numerical data used to generate Figure 4 and its supplement. Figure 5-source data 1, Figure 5-figure supplement 1-source data 1, Figure 5-figure supplement 2-source data 1, and Figure 5-figure supplement 3-source data 1 contain the numerical data used to generate Figure 5 and its supplements. Figure 7-source data 1, Figure 7-figure supplement 2-source data 1, Figure 7-figure supplement 3-source data 1, and Figure 7-figure supplement 4-source data 1 contain the numerical data used to generate Figure 7 and its supplements. All custom analysis scripts, DANCE classifiers, and documentation used in this study are publicly available in GitHub repository at https://github.com/agrawallab/DANCE (copy archived at _Dey and Agrawal, 2025_).

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
