## [Editor Report · eLife Assessment]

This study presents a **valuable** open-source and cost-effective method for automating the quantification of male aggression and courtship in *Drosophila melanogaster*. The work as presented provides **solid** evidence that the use of the behavioral setup that the authors designed - using readily available laboratory equipment and standardised high-performing classifiers they developed using existing software packages - accurately and reliably characterises social behavior in Drosophila. The work will be of interest to Drosophila neurobiologists and particularly to those working on male social behaviors.

---

## [Referee Report · Reviewer #1 (Public review)]

The study introduces an open-source, cost-effective method for automating the quantification of male social behaviors in *Drosophila melanogaster*. It combines machine-learning based behavioral classifiers developed using JAABA (Janelia Automatic Animal Behavior Annotator) with inexpensive hardware constructed from off-the-shelf components. This approach addresses the limitations of existing methods, which often require expensive hardware and specialized setups. The authors demonstrate that their new "DANCE" classifiers accurately identify aggression (lunges) and courtship behaviors (wing extension, following, circling, attempted copulation, and copulation), closely matching manually annotated ground-truth data. Furthermore, DANCE classifiers outperform existing rule-based methods in accuracy. Finally, the study shows that DANCE classifiers perform as well when used with low-cost experimental hardware as with standard experimental setups across multiple paradigms, including RNAi knockdown of the neuropeptide Dsk and optogenetic silencing of dopaminergic neurons.

The authors make creative use of existing resources and technology to develop an inexpensive, flexible, and robust experimental tool for the quantitative analysis of Drosophila behavior. A key strength of this work is the thorough benchmarking of both the behavioral classifiers and the experimental hardware against existing methods. In particular, the direct comparison of their low-cost experimental system with established systems across different experimental paradigms is compelling. A weakness of the study is that the use of JAABA-based classifiers to analyze aggression and courtship is not novel (Tao et al., J. Neurosci., 2024; Sten et al., Cell, 2023; Chiu et al., Cell, 2021; Isshi et al., eLife, 2020; Duistermars et al., Neuron, 2018). However, the demonstration the JAABA classifiers they developed work as well without expensive experimental hardware opens the door to more low-cost systems for quantitative behavior analysis.

In summary, this work provides a practical and accessible approach to quantifying Drosophila behavior, reducing the economic barriers to the study of the neural and molecular mechanisms underlying social behavior.

---

## [Referee Report · Reviewer #2 (Public review)]

Summary:

This manuscript addresses the development of a low-cost behavioural setup and standardised open-source high performing classifiers for aggression and courtship behaviour. It does so by using readily available laboratory equipment and previously developed software packages. By comparing the performance of the setup and the classifiers to previously developed ones, this study shows the classifier's overperformance and the reliability of the low-cost setup in recapitulating previously described effects of different manipulations on aggression and courtship.

Strengths:

The newly developed classifiers for lunges, wing extension, attempted copulation, copulation, following, circling, perform better than previously available developed ones. The behavioural setup developed is low cost and reliably allows analysis of both aggression and courtship behaviour, validated through social experience manipulation (social isolation), gene knock (Dsk in Dilp2 neurons) and neuronal inactivation (dopaminergic neurons) know to affect courtship and aggression.

Weaknesses:

This framework only encompasses analysis of lunges, while aggression encompasses multiple behaviours. Even though DANCE can serve as a template allowing future development of additional classifiers, the current study compares performance to CADABRA which analyses further aggression behaviours, making the comparisons incomplete.

---

## [Referee Report · Reviewer #3 (Public review)]

The study by Yadav et al. describes a new setup to quantify a number of aggression and mating behaviors in *Drosophila melanogaster*. The investigation of these behaviors requires the analysis of large number of videos to identify each kind of behavior displayed by a fly. Several approaches to automatize this process have been published before, but each of them has their limitations. The authors set out to develop a new setup that includes a very low-cost, easy to acquire hardware and open-source machine-learning classifiers to identify and quantify the behavior.

Strengths:

(1) The study demonstrates that their cheap, simple, and easy to obtain hardware works just as well as custom-made, specialized hardware for analyzing aggression and mating behavior. This enables the setup to be used in a wide range of settings, from research with limited resources to classroom teaching.

(2) The authors used previously published software to train new classifiers for detecting a range of behaviors related to aggression and mating and make them freely available. The classifiers are very positively benchmarked against a manually acquired ground-truth as well as existing algorithms.

(3) The study demonstrates the applicability of the setup (hardware and classifiers) to common methods in the field by confirming a number of expected phenotypes with their setup.

Taken together, this work can greatly facilitate research of aggression and mating in Drosophila. The combination of low-cost, off-the-shelf hardware and open-source, robust software enables researchers with very little funding or technical expertise to contribute to the scientific process, and also allows large-scale experiments, for example, in classroom teaching with many students, or for systematic screenings.

---

## [Author Response]

The following is the authors’ response to the original reviews.

**Reviewer #1 (Public review):**
The study introduces an open-source, cost-effective method for automating the quantification of male social behaviors in *Drosophila melanogaster*. It combines machine-learning-based behavioral classifiers developed using JAABA (Janelia Automatic Animal Behavior Annotator) with inexpensive hardware constructed from off-the-shelf components. This approach addresses the limitations of existing methods, which often require expensive hardware and specialized setups. The authors demonstrate that their new "DANCE" classifiers accurately identify aggression (lunges) and courtship behaviors (wing extension, following, circling, attempted copulation, and copulation), closely matching manually annotated groundtruth data. Furthermore, DANCE classifiers outperform existing rule-based methods in accuracy. Finally, the study shows that DANCE classifiers perform as well when used with low-cost experimental hardware as with standard experimental setups across multiple paradigms, including RNAi knockdown of the neuropeptide Dsk and optogenetic silencing of dopaminergic neurons.The authors make creative use of existing resources and technology to develop an inexpensive, flexible, and robust experimental tool for the quantitative analysis of Drosophila behavior. A key strength of this work is the thorough benchmarking of both the behavioral classifiers and the experimental hardware against existing methods. In particular, the direct comparison of their low-cost experimental system with established systems across different experimental paradigms is compelling.While JAABA-based classifiers have been previously used to analyze aggression and courtship (Tao et al., J. Neurosci., 2024; Sten et al., Cell, 2023; Chiu et al., Cell, 2021; Isshi et al., eLife, 2020; Duistermars et al., Neuron, 2018), the demonstration that they work as well without expensive experimental hardware opens the door to more low-cost systems for quantitative behavior analysis.

We thank the reviewer for their positive assessment and constructive suggestions. We have cited these additional JAABA studies in the Introduction**.** We clarified that several prior JAABA-based classifiers were developed using specialized machinevision cameras or custom setups, and that in some cases the original code and classifiers were not made publicly available, which limits reproducibility and wider adoption. To address this, we explicitly note in the revised manuscript that DANCE was developed with accessibility in mind.

Although the study provides a detailed evaluation of DANCE classifier performance, its conclusions would be strengthened by a more comprehensive analysis. The authors assess classifier accuracy using a bout-level comparison rather than a frame-level analysis, as employed in previous studies (Kabra et al., Nat Methods, 2013). They define a true positive as any instance where a DANCE-detected bout overlaps with a manually annotated ground-truth bout by at least one frame. This criterion may inflate true positive rates and underestimate false positives, particularly for longer-duration courtship behaviors. For example, a 15-second DANCE-classified wing extension bout that overlaps with ground truth for only one frame would still be considered a true positive. A frame-level analysis performance would help address this possibility.

We thank the reviewer for raising this important point. Our original use of bout-level analysis followed existing literature (Duistermars et al., 2018; Ishii et al., 2020; Chiu et al., 2021; Tao et al., 2024; Hindmarsh Sten et al., 2025). While our lunge classifier already operates at the frame level, we have now performed additional frame-level evaluations for the duration based courtship classifiers. These analyses revealed only minor differences in precision, recall, and F1 scores compared with the original bout-level approach (see new Figure 5—Figure Supplement 3). Details of this analysis are now included in the Materials and Methods.

In summary, this work provides a practical and accessible approach to quantifying Drosophila behavior, reducing the economic barriers to the study of the neural and molecular mechanisms underlying social behavior.

We thank the reviewer for their encouraging comments and for recognizing the accessibility and practical value of our approach. We appreciate the constructive suggestions, which have helped strengthen the manuscript.

**Reviewer #2 (Public review):**
Summary:This manuscript addresses the development of a low-cost behavioural setup and standardised open-source high-performing classifiers for aggression and courtship behaviour. It does so by using readily available laboratory equipment and previously developed software packages. By comparing the performance of the setup and the classifiers to previously developed ones, this study shows the classifier's overperformance and the reliability of the low-cost setup in recapitulating previously described effects of different manipulations on aggression and courtship.Strengths:The newly developed classifiers for lunges, wing extension, attempted copulation, copulation, following, and circling, perform better than available previously developed ones. The behavioural setup developed is low cost and reliably allows analysis of both aggression and courtship behaviour, validated through social experience manipulation (social isolation), gene knock (Dsk in Dilp2 neurons) and neuronal inactivation (dopaminergic neurons) known to affect courtship and aggression.

We thank the reviewer for the clear summary of our work and for highlighting its strengths. We appreciate these positive comments and suggestions, which have helped improve the clarity of the manuscript.

Weaknesses:Aggression encompasses multiple defined behaviours, yet only lunges were analysed. Moreover, the CADABRA software to which DANCE was compared analyses further aggression behaviours, making their comparisons incomplete. In addition, though DANCE performs better than CADABRA and Divider in classifying lunges in the behavioural setup tested, it did not yield very high recall and F1 scores.

We thank the reviewer for raising this important point. We focused on lunges because they are widely used as a standard proxy for male aggression across multiple laboratories (Agrawal et al., 2020; Asahina et al., 2014; Chiu et al., 2021; Chowdhury et al., 2021; Dierick et al., 2007; Hoyer et al., 2008; Jung et al., 2020; Nilsen et al., 2004; Watanabe et al., 2017). As noted in the Discussion, our study also provides a template for the future development of additional aggression classifiers (fencing, wing flick, tussle, chase, female headbutt) and courtship classifiers (tapping, licking, rejection), which can be trained and shared through the same DANCE framework. Developing and validating these was beyond the scope of the present work.

To address the concern regarding precision, recall, and F1 scores, we performed additional analyses across all training videos and compiled these results in the new Figure 2—Figure Supplement 2. Our earlier lunge classifier had performance metrics obtained after training on a total of 11 videos. Our analysis shows performance metrics for classifiers trained on four independent datasets (Videos 8– 11). We found that the classifier trained on nine videos provided the best balance of precision, recall, and F1 (78.73%, 73.07%, and 75.79%, respectively), which was slightly better than the earlier classifier. We therefore updated the main figure, text, and Materials and Methods to use this version and uploaded the corresponding classifier and training details to the GitHub repository.

DANCE is of limited use for neuronal circuit-level enquiries, since mechanisms for intensity and temporally controlled optogenetic manipulations, which are nowadays possible with open-source software and low-cost hardware, were not embedded in its development.

We thank the reviewer for this valuable point. The primary aim of DANCE is to provide an accessible, modular, and low-cost behavioural recording and analysis platform. It was designed so that users can readily integrate additional components such as optogenetic control when needed. As a proof of concept, we implemented optogenetic silencing of dopaminergic neurons using the DANCE hardware and confirmed that this manipulation increased aggression (Figure 7R).

To facilitate adoption, we now provide schematic diagrams, LED control code, and instructions on our GitHub page and setup photographs in the manuscript (see new Figure 7—Figure Supplement 1). The released code allows programmable timing and intensity control, enabling users to reproduce temporally precise optogenetic protocols or extend the system for other stimulation paradigms.

**Reviewer #3 (Public review):**
The preprint by Yadav et al. describes a new setup to quantify a number of aggression and mating behaviors in *Drosophila melanogaster*. The investigation of these behaviors requires the analysis of a large number of videos to identify each kind of behavior displayed by a fly. Several approaches to automatize this process have been published before, but each of them has its limitations. The authors set out to develop a new setup that includes very low-cost, easy-to-acquire hardware and open-source machine-learning classifiers to identify and quantify the behavior.Strengths:(1) The study demonstrates that their cheap, simple, and easy-to-obtain hardware works just as well as custom-made, specialized hardware for analyzing aggression and mating behavior. This enables the setup to be used in a wide range of settings, from research with limited resources to classroom teaching.(2) The authors used previously published software to train new classifiers for detecting a range of behaviors related to aggression and mating and to make them freely available. The classifiers are very positively benchmarked against a manually acquired ground truth as well as existing algorithms.(3) The study demonstrates the applicability of the setup (hardware and classifiers) to common methods in the field by confirming a number of expected phenotypes with their setup.

We thank the reviewer for the positive assessment of our work and for highlighting its strengths. We appreciate these encouraging comments and suggestions, which have helped improve the clarity and presentation of the manuscript.

Weaknesses:(1) When measuring the performance of the duration-based classifiers, the authors count any bout of behavior as true positive if it overlaps with a ground-truth positive for only 1 frame - despite the minimal duration of a bout is 10 frames, and most bouts are much longer. That way, true positives could contain cases that are almost totally wrong as long there was an overlap of a single frame. For the mating behaviors that are classified in ongoing bouts, I think performance should be evaluated based on the % of correctly classified frames, not bouts.

We thank the reviewer for raising this concern. In response to this point, and to Reviewer #1’s similar comment, we performed a frame-level evaluation of all duration-based courtship classifiers. The analysis revealed only minor differences compared with the original bout-level metrics (see new Figure 5—Figure Supplement 3), confirming the robustness of our classifiers. We have also added a description of this analysis in the Materials and Methods section.

(2) In the methods part, only one of the pre-existing algorithms (MateBook), is described. Given that the comparison with those algorithms is a so central part of the manuscript, each of them should be briefly explained and the settings used in this study should be described.

We thank the reviewer for this helpful suggestion. In the revised manuscript, we expanded the Materials and Methods to include concise descriptions and parameter settings for all pre-existing algorithms used for comparison. This includes dedicated subsections for CADABRA and the Divider assay, with explicit reference to their rulebased or geometric features. For MateBook, we specified the persistence filters used and the adjustments made for fair benchmarking. These changes ensure transparency and reproducibility.

Taken together, this work can greatly facilitate research on aggression and mating in Drosophila. The combination of low-cost, off-the-shelf hardware and open-source, robust software enables researchers with very little funding or technical expertise to contribute to the scientific process and also allows large-scale experiments, for example in classroom teaching with many students, or for systematic screenings.

We thank the reviewer for the encouraging comments and for recognizing the accessibility and broad applicability of DANCE. We believe these revisions have further strengthened the manuscript.

**Reviewer #1 (Recommendations for the authors):**
The following comments highlight areas where additional context, clarification, or further analysis could strengthen the manuscript. I hope these suggestions will be useful in refining your work.(1) Lines 71-73: The authors state that Ctrax "leads to frequent identity switches among tracked flies, which is not the case while using FlyTracker." However, Ctrax was specifically designed to minimize identity errors, and Kabra et al. (2013) reported a low frequency of such errors-approximately one per five fly-hours in 10-fly videos. In contrast, Caltech FlyTracker does not correct identity errors automatically, requiring manual corrections, as noted in the Methods section of this study. If this is not an oversight, please provide further context to clarify this distinction.

We thank the reviewer for raising this clarification. As reported by Bentzur et al. (2021), when groups of flies were tracked simultaneously, Ctrax often generated multiple identities for the same individual, sometimes producing more trajectories than the actual number of flies. To prevent ambiguity, we revised the text to read: “While both Ctrax and FlyTracker (Eyjolfsdottir et al., 2014) may produce identity switches, when groups of flies were tracked simultaneously, Ctrax led to inaccuracies that required manual correction using specialized algorithms such as FixTrax (Bentzur et al., 2021).” We also quantified FlyTracker identity-switch rates in our datasets and report them in new Supplementary File 5, confirming that such events were rare (< 2% of tracked intervals). We believe, this updated version provides the necessary context and ensures accuracy in describing each tracker’s limitations.

(2) Line 85: Providing additional context on how this study builds on previous work using JAABA-based classifiers for fly social behavior and comparing these classifiers to rule-based methods would more accurately situate it within the field. The authors state that "recently, a few JAABA-based classifiers have been developed for measuring aggression and courtship" and cite four related studies. However, this statement seems to underrepresent the use of JAABA-based classifiers for quantifying fly social behavior, which has become common in the field. Several additional studies (as noted in the public review) have developed JAABA-based classifiers for scoring aggression or courtship. Furthermore, other studies have compared the performance of JAABA-based classifiers with rule-based classifiers like CADABRA (e.g., Chowdhury et al., Comm Biology 2021; Leng et al., PlosOne 2020; Kabra et al., Nat Methods 2013). Mentioning the similar findings in those studies and your own helps strengthen the conclusion that machine-learning-based classifiers outperform rule-based classifiers in several experimental contexts.

We thank the reviewer for this helpful suggestion. We have revised the Introduction to include additional references to studies that applied JAABA-based classifiers for aggression and courtship and made textual edits to reflect this. We further noted that, unlike several previous studies, all DANCE classifiers and analysis code are publicly available.

**Reviewer #2 (Recommendations for the authors):**
(1) Suggestions for improved or additional experiments, data or analyses: As mentioned in the description of the effect of optogenetic inactivation of dopaminergic neurons, in the conclusion and also reported in the literature, there are other important identified aggression behaviours, such as fencing, wing flick, tussle, and chase. Similarly, for courtship, tapping and licking have also been defined. This study, as opposed to proposed future studies, would benefit from creating opensource classifiers for these established behaviours, which are important for the analysis of aggression and courtship.

We thank the reviewer for this valuable suggestion. As clarified in the Discussion, this manuscript intentionally focuses on six core, well-validated aggression and courtship behaviors to demonstrate DANCE’s modularity and reproducibility. Developing additional classifiers such as fencing, wing flick, tussle, chase, tapping, and licking would require extensive annotation and validation beyond the present scope. To address this point, we explicitly note in the revised text that the DANCE pipeline is readily extendable, allowing the community to build new classifiers within the same framework.

In terms of observer bias assessment for ground-truthing in courtship, this was only presented for circling and it would be beneficial to have encompassed all behaviours analysed.

We thank the reviewer for this suggestion. Observer-bias comparisons for all six classifiers are presented in Figure 2—Figure Supplement 1 (panels A–F). We clarified in the Results that annotations from two independent evaluators were compared for all classifiers, with no significant differences observed, confirming their robustness.

Finally, intensity and temporal optogenetic control are important for neuronal circuit analysis of underlying behaviour. The authors could embed this aspect in DANCE by integrating control of the green light LED strip used in this study using, for example, the open-source visual reactive programming software Bonsai (Lopes et al., 2015) and open-source electronics platform Arduino. This is an important and valuable addition in line with maintaining low cost.

We thank the reviewer for this valuable suggestion. DANCE was designed to be modular, allowing integration of temporal optogenetic control. To support immediate adoption, we now provide Arduino LED control code, setup schematics, and photographs (new Figure 7—Figure Supplement 1) along with step-by-step instructions on our GitHub page. We also note that Bonsai and Arduino frameworks are compatible with DANCE, enabling future extensions for closed-loop or behaviortriggered stimulation.

(2) Minor corrections to the text and figures:Figure Supplement 1 refers only to Figure 2, yet panels D-F refer to the behaviour circling in courtship and therefore should be assigned to the respective figure.

Thanks, we have corrected this.

In lines 315-316, the cumbersome task of fluon coating for aggression assays seems to be ubiquitous across assays which is not the case, and therefore the sentence should include the word 'some'.

Thanks, we have edited this.

The cost of the phone and/or tablet should be included in the DANCE setup costs, as presumably these devices will be dedicated to the behavioural studies, for consistency purposes.

We thank the reviewer for this comment. We intentionally did not include smartphones or tablets in the setup cost because, in our experiments, these devices were not dedicated exclusively to DANCE but were repurposed from routine personal use. Our aim was to leverage readily available consumer electronics so that their cost does not become a barrier to adoption. We confirmed that commonly available Android phones capable of 30 fps at 1080p in H.264 format, as well as tablets or phones running a simple white-screen light app, are sufficient for reliable behavior classification and illumination. Since these devices can be returned to regular use after recordings, including their cost in the setup would not accurately reflect the intended accessibility of DANCE. For consistency, we now clarify in the Materials and Methods that such devices should be placed in airplane mode during recordings.

**Reviewer #3 (Recommendations for the authors):**
(1) For my taste, the authors put too much emphasis on the point that their method outperforms existing methods. I understand the value in comparing to published methods and it is of course fully justified to state the advantages of the new method. But the whole preprint is set up as a competition with the old algorithms, and the conclusion that the new classifier is better is repeated in each figure caption and after each paragraph of the results. This competitive mindset also extends to the selection of which results are presented as main figures and which as supplements - all cases in which the previous methods actually perform well are only presented in the supplement. I think this is simply unnecessary as the authors' results speak for themselves, and do not need the continuous competitive comparison.

We thank the reviewer for this thoughtful suggestion. Our intention was to benchmark DANCE rigorously against existing methods, not to frame the study competitively. We agree that repeated emphasis on relative performance was unnecessary. In the revised version, we streamlined figure captions and text throughout the manuscript to balance comparisons and removed redundant phrasing. Instances where other methods performed well are now presented with equal clarity to maintain a neutral and informative tone.

(2) When describing the DANCE hardware, as a reader I would find it interesting to also read about potential issues that the authors encountered. For example, how difficult is it to handle the materials without breaking or deforming them, which could affect the behavioral assays? How critical is it to use specific blister packs - the availability of which will likely vary strongly between countries? Did the authors try different sizes, and products? Such information, even as a supplement, could be very helpful for the widespread use of the hardware.

We thank the reviewer for this important point. To address this, we conducted additional tests comparing DANCE arenas of different diameters (new Figure 7— Figure Supplement 3A–C and new Figure 7—Figure Supplement 4A–L). We also consulted colleagues in multiple countries and verified that the blister packs used in our assays are readily available. The Materials and Methods now include practical handling notes: blister foils can be reused ~30–40 times for aggression assays and ~10–15 times for courtship assays before deformation. We also describe how to prevent agar surface damage during assembly and how to wash and dry the arenas for optimal reusability.

(3) I find the arrows pointing to several videos in a number of figures rather distracting and redundant, and suggest omitting them.

Thanks**,** we have omitted these arrows from all relevant figures and clarified the figure legends to enhance readability.

(4) P8, line 169 ff: this is a very long sentence that should be separated into several sentences.

We have rewritten this as follows: “DANCE scores remained comparable to groundtruth scores across all categories, whereas CADABRA and Divider underestimated the lunge counts (Figure 2B–E). Correlation analysis revealed a strong relationship between DANCE and ground-truth scores (Figure 2F, Supplementary File 2). In comparison, CADABRA and the Divider assay classifier showed a weaker correlation (Figure 2G-H, Supplementary File 2).”

(5) P10, line 216: please explain, here and in the methods, how these behavioral indices are calculated. I did not find this information anywhere in the paper.

We thank the reviewer for pointing this out. We now define the behavioral index explicitly in Materials and Methods: “For each assay, a behavioral index was calculated as the proportion of frames in which the male engaged in the specified behavior. This was obtained by dividing the total number of frames annotated for that behavior by the total number of frames in the recording.”

(6) P11, line 253: I don't understand the modifications to MateBook regarding attempted copulations, neither in the results nor the methods section. I would ask the authors to explain more explicitly what was done.

We thank the reviewer for this helpful suggestion. We have re-written several parts of the Materials and methods to clarify these details and streamline the text. To train the attempted copulation classifier, we combined datasets from assays with mated and decapitated virgin females, using manual annotations as ground truth. We also adapted MateBook’s persistence filters (Ribeiro et al., 2018) and defined thresholds explicitly: mounting lasting >45 s (>1350 frames at 30 fps) was defined as copulation, whereas abdominal curling without mounting, or mounting lasting 0.33– 45 s, was defined as attempted copulation.

(7) Figure 7F: this is the only case with a significant difference between the two setups. What explanations do the authors have for the discrepancy?

We thank the reviewer for raising this point**.** After repeating the experiments, we no longer found a significant difference between the setups. Figure 7 and its legend have been updated to reflect these results.

(8) Figure 2 - Supplement 1: I do not understand why the boxes for Observer 1 have different colors in different figures. Does this have a meaning?

Thanks for pointing this out. The color differences had no intended meaning, and we have corrected the figure for consistency across panels.

(9) P22, line 517ff: It would be interesting to know how frequently identity switches occurred. For large-scale, automatic behavioral screenings that step could be a crucial bottleneck.

We thank the reviewer for this valuable suggestion. We analyzed identity switches using the FlyTracker “Visualizer” package, which flags frames with possible overlaps or jumps. Flagged intervals were manually verified, and we report these data in new Supplementary File 5. Identity switch rates were very low: 0.66% for high-resolution recordings and 1.9% for smartphone DANCE videos in the most challenging decapitated-virgin dataset. These findings demonstrate robust tracking performance under both setups.